# Physical and optical characteristics of heavily melted "rotten" Arctic sea ice

C. M. Frantz[1,2], B. Light[1], S. M. Farley[1], S. Carpenter[1], R. Lieblappen[3,4], Z. Courville[4], M. V. Orellana[1,5], and K. Junge[1]

[1]Polar Science Center, Applied Physics Laboratory, University of Washington, Seattle, Washington, 98105 USA

[2]Department of Geosciences, Weber State University, Ogden, Utah, 84403 USA

[3]Vermont Technical College, Randolph, VT, 05061 USA

[4]US Army Engineer Research Development Center - Cold Regions Research and Engineering Laboratory, Hanover, New Hampshire, 39180 USA

[5]Institute for Systems Biology, Seattle, Washington, 98109 USA

*Correspondence to*: Bonnie Light (bonnie@apl.washington.edu)

**Abstract.** Field investigations of the properties of heavily melted "rotten" Arctic sea ice were carried out on shorefast and
drifting ice off the coast of Utqiaġvik (formerly Barrow), Alaska during the melt season. While no formal criteria exist to
qualify when ice becomes "rotten", the objective of this study was to sample melting ice at the point where its structural and
optical properties are sufficiently advanced beyond the peak of the summer season. Baseline data on the physical
(temperature, salinity, density, microstructure) and optical (light scattering) properties of shorefast ice were recorded in May
and June 2015. In July of both 2015 and 2017, small boats were used to access drifting "rotten" ice within ~32 km of
Utqiaġvik. Measurements showed that pore space increased as ice temperature increased (-8 °C to 0 °C), ice salinity
decreased (10 ppt to 0 ppt), and bulk density decreased (0.9 g cm$^{-3}$ to 0.6 g cm$^{-3}$). Changes in pore space were characterized
with thin-section microphotography and X-ray micro-computed tomography in the laboratory. These analyses yielded
changes in average brine inclusion number density (which decreased from 32 mm$^{-3}$ to 0.01 mm$^{-3}$), mean pore size (which
increased from 80 μm to 3 mm) as well as total porosity (increased from 0% to > 45%) and structural anisotropy (variable,
with values generally less than 0.7). Additionally, light scattering coefficients of the ice increased from approximately 0.06
cm$^{-1}$ to > 0.35 cm$^{-1}$ as the ice melt progressed. Together, these findings indicate that the properties of Arctic sea ice at the
end of melt season are significantly distinct from those of often-studied summertime ice. If such rotten ice were to become
more prevalent in a warmer Arctic with longer melt seasons, this could have implications for the exchange of fluid and heat
at the ocean surface.

**1 Introduction**

The seasonal evolution of Arctic sea ice follows a fairly predictable annual pattern: winter, snow melt, pond formation, pond
drainage, rotten ice [*DeAbreu et al.*, 2001]. Considerable attention has been given to characterization of these various states
and their transitions. *In situ* observations during the summer melt season are typically straightforward through the pond
drainage stage, but, as ice conditions deteriorate, it becomes increasingly difficult to work on or around the most fragile
state, rotten ice. During the summer of 1894, Nansen, in his seminal work *Farthest North* (1897, p. 433) described it well,
"Everything is in a state of disintegration, and one's foothold gives way at every step." Extensive areas of rotten ice in the
Beaufort Sea pack were encountered in September 2009 [*Barber et al.* 2009], where the ice cover was found to be composed
of small remnants of decayed and drained ice floes interspersed with new ice. The remotely sensed radiometric
characteristics of this ice cover appeared indistinguishable from old, thick multiyear ice. Such characterization is largely
indicative of the physical properties of the ice on meter to decameter scales, but the microstructural properties of melting sea
ice at the very end of its summer melt remain largely undocumented.
The relatively high temperatures and abundant sunlight of summer cause sea ice to "rot". While the microstructure of winter
ice is characterized by small, isolated brine inclusions, with brine convection restricted to the lower reaches of the ice, and
spring ice is characterized by increased permeability and brine convection through the full depth of the ice cover [*Jardon et*
*al.*, 2013; *Zhou et al.*, 2013], the defining characteristics of rotten ice may be its high porosity and enhanced permeability.
Warming causes changes in the ice structure including enlarged and merged brine and gas inclusions (see, e.g., *Weeks and*
*Ackley*, 1986; *Light et al.*, 2003). Columnar ice permeability increases drastically for fluid transport when the brine volume
fraction exceeds approximately 5% [*Golden et al.*, 2007; *Pringle et al.*, 2009]. In a previous study on shorefast ice, brine
volume fractions were found to exceed this 5% threshold for permeability through the entire depth of the ice from early May
onwards [*Zhou et al.*, 2013]. While the term "rotten ice" is used in this manuscript to refer to heavily melted summer ice that
has diminished structural integrity, relatively large voids, and is highly permeable, it is also noted that this work is intended
to provide a more refined and quantitative definition of this ice type.
Connectivity of the pore space in sea ice is known to contribute to ocean-atmosphere heat transfer [*Weeks and Ackley*, 1986;
*Hudier et al.*, 1995; *Lytle and Ackley*, 1996; *Weeks*, 1998; *Eicken et al.*, 2002], exchange of dissolved and particulate matter
[*Freitag*, 1999; *Krembs et al.*, 2000] including nutrients [*Fritsen et al.*, 1994], salinity evolution of the ice cover
[*Untersteiner*, 1968; *Wettlaufer et al.*, 2000; *Vancoppenolle et al.*, 2007], and surface melt pond distribution [*Eicken et al.*,
2002]. As a result of this notable connectivity, rotten ice also has reduced structural integrity, which can have implications
for ice dynamics. Though it is known to have diminished tensile and flexural strength [*Richter-Menge and Jones*, 1993;
*Timco and O'Brien*, 1994; *Timco and Johnston*, 2002], such details have not been well-characterized. Measurements by
*Timco and Johnston* [2002] demonstrated that in mid-May, the ice had about 70% of its mid-winter strength. By early June,
about 50% and by the end of June, 15%–20% of its mid-winter strength. The ice strength during July was only about 10% of
midwinter strength. Such changes in strength may be relevant to the late summer behavior of Arctic ice-obligate megafauna.
With increasing melt season length [*Stroeve et al.*, 2014], the future could bring increasing areas of rotten ice. Because it
represents the very end of summer melt, its presence matters for the longevity of the ice cover. If the ice melts completely,
then the open ocean will form new ice in the autumn. Only ice remaining at the end of summer can become second-year, and
subsequently, multiyear ice.
For rotten ice, permeability is typically large enough to render the ice cover to be in connection with the ocean throughout its
depth. As a result, rotten ice may have a very different biogeochemical environment for sea-ice microbial communities than
ice with connectivity properties typical of winter, spring, or even early to mid-summer. Increases in ice permeability result in
an increase in the flow rate of surface meltwater that can penetrate through a melting ice cover, both from the top of the ice
downwards [e.g., *Untersteiner*, 1968], as well as from the bottom of the ice upwards [e.g., *Eicken et al.*, 2002; *Jardon et al.*,
2013]. The convective overturning of meltwater pooled beneath the ice can contribute significantly to enlargement of pores
and internal melt. In fact, during the Surface Heat Budget of the Arctic Ocean (SHEBA) field campaign, *Eicken et al.* [2002]
noted that high advective heat fluxes into the permeable ice found on melt pond bottoms and first-year ice likely contributed
to the breakup and disintegration of the ice cover toward the end of the melt season.
To address questions about the physical characteristics of rotten sea ice, a targeted field study was carried out at Utqiaġvik
(formerly Barrow; 71.2906° N, 156.7886° W), Alaska during May, June, July 2015, with further sample collection carried
out in July 2017. The May and June sampling sessions were for the purpose of collecting ice to be used for baseline studies
and were carried out on landfast ice. In July, small boats were used to search for, and sample, rotten ice off the coast.

## 2 Materials and methods

### 2.1 Sample collection and description

Sea ice samples and field measurements were collected from locations near the north coast of Alaska (Fig. 1-2, Table A1).
Samples were collected at three different time points to help define the progression of melt: May to collect baseline data on
the ice properties, June to observe its progression, and July to capture rotten ice (Fig. 3).
Locations sampled and cores collected are summarized in Table 1. All ice cores were drilled using a 9-cm diameter Kovacs
Mark II corer (Kovacs Enterprise, Roseburg, Oregon, USA) through the full depth of the ice. Extracted cores were
photographed and either bagged whole or as 20-cm subsections for subsequent laboratory analysis. At each sampling site, a
single core was used for temperature and density profiles. Bagged cores were stored up to several hours in insulated coolers
for transport back to the Barrow Arctic Research Center (BARC) laboratory, and immediately placed in one of several walk-
in freezers set to -20 °C for archival cores to be saved for later processing, or, for cores processed at BARC, at approximate
average *in situ* core temperatures (-5 °C in May, -2 °C in June, -1 °C in July), referred to subsequently in this text as
"working" temperatures.

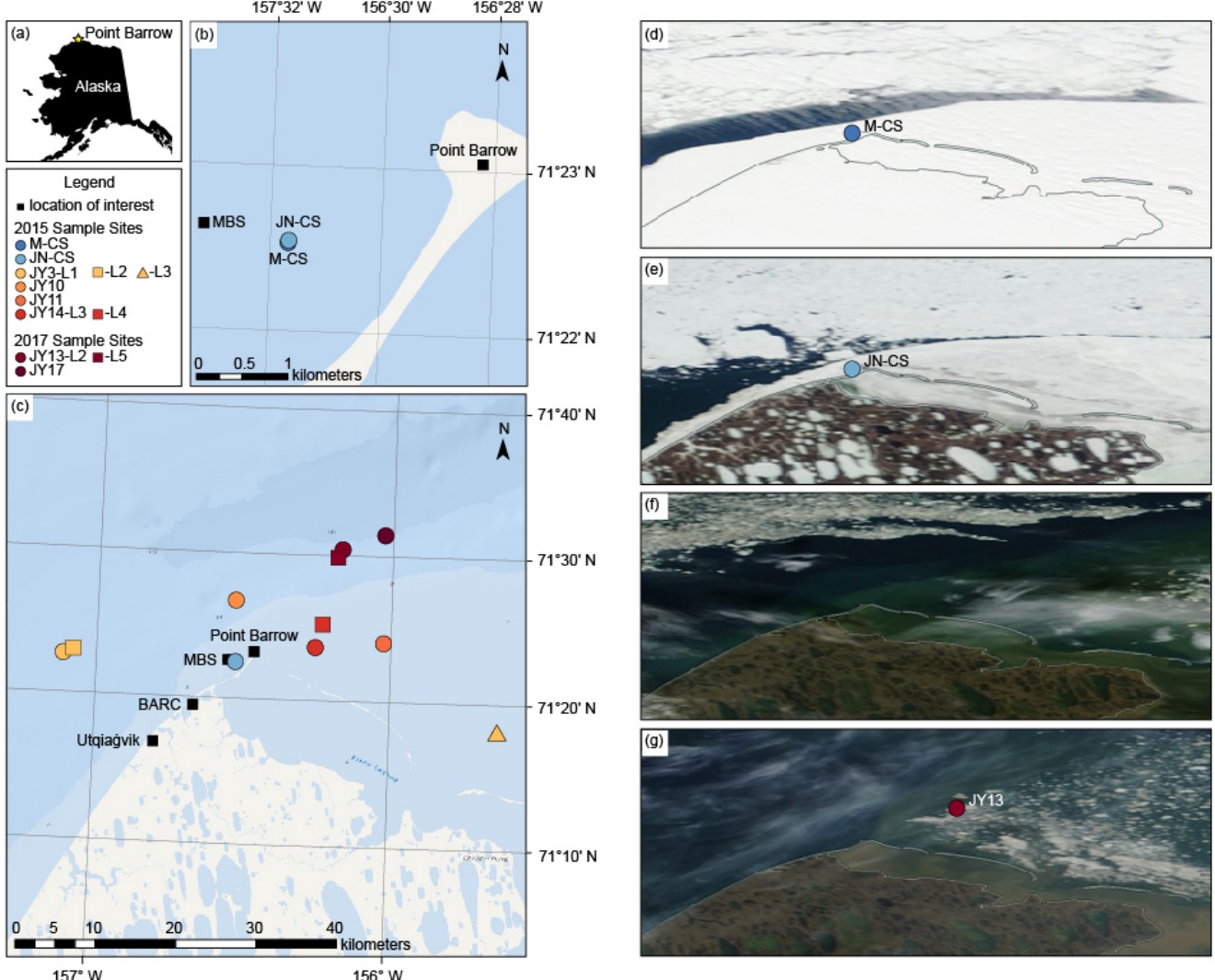

Figure 1. Map of sea ice sample collection sites. (a) Point Barrow (star). (b) Landfast sea ice sample collection sites for May
2015 (M-CS, dark blue) and June 2015 (JN-CS, light blue), shown relative to the 2015 SIZONet Mass Balance Site (MBS)
and Point Barrow. M-CS and JN-CS were separated by less than 30 m. (c) Ice sample collection sites in May 2015 (dark
blue), June 2015 (light blue), July 2015 (orange and red), and July 2017 (magenta) relative to Point Barrow, the 2015 MBS,
the Barrow Arctic Research Center (BARC), and the town of Utqiaġvik, Alaska, USA. Alaska Coastline base map provided
by the Alaska Department of Natural Resources (1998). ArcGIS Ocean base map sources: Esri, GEBCO, NOAA, National
Geographic, DeLorme, HERE, Geonames.org, and other contributors (2016). (d-g) NASA MODIS satellite images of Point
Barrow on clear-sky days on (d) 7 May 2015, showing the location of the M-CS sample site (dark blue); (e) 7 June 2015,
showing the location of the JN-CS sample site (light blue); (f) 6 July 2015, showing the general locations of highly mobile
ice proximal to Point Barrow (cloud cover obscures the region during the days samples were collected in July 2015); and (g)
13 July 2017, showing the location of the JY-13 sample site (magenta). No clear-sky images were available for the July 2015
sampling dates (JY3, JY10, JY11, and JY14) or for 17 July 2017 (JY17). Satellite imagery retrieved from
worldview.earthdata.nasa.gov (2017), coastline overlay © OpenStreetMap contributors, available under the Open Database
License.

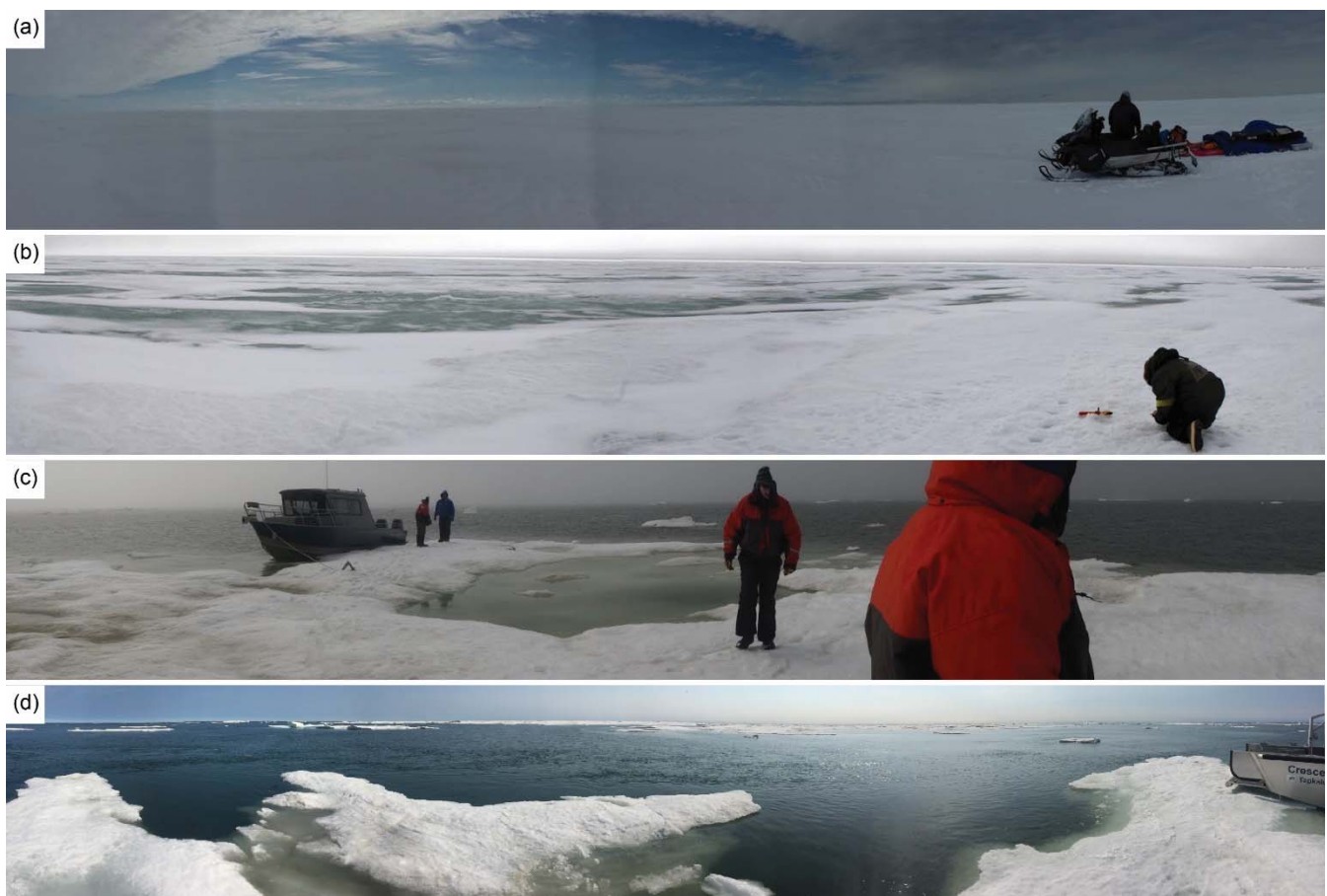

Figure 2. Sea ice in the vicinity of Utqiaġvik, Alaska, USA during summer melt. a) Photomosaic of 8 May 2015 sample site
(M-CS). b) Panorama photograph of 7 June 2015 sample site (JN-CS). c) Panorama photograph of the floe sampled on 11
July 2015 (JY-11). d) Panorama photograph of floe L2 sampled on 13 July 2017 (JY13-L2).

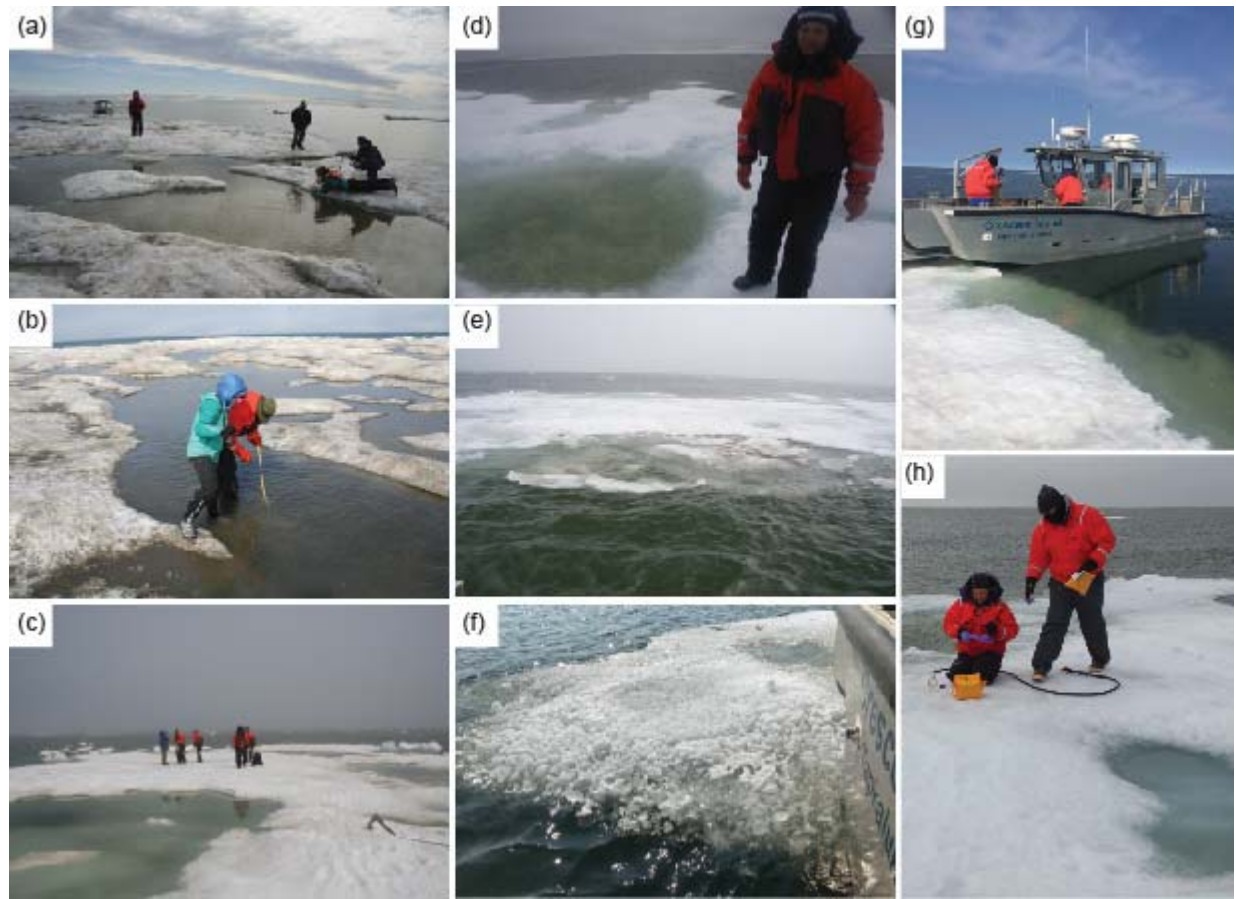

Figure 3. Photographs of specific sampling sites in July: (a) JY3-L3, (b) JY10, (c) JY11, (d) JY14-L3, (e) JY14-L4, (f) JY13-L1, (g) JY13-L2, (h) JY17-L1.

### 2.1.1 May 2015

The first set of samples was collected on 6 May 2015 from landfast, first-year, snow-covered congelation sea ice in a region of undeformed ice 2 km southwest of Point Barrow and ~0.9 km due east of the University of Alaska Fairbanks 2015 SIZONet Sea Ice Mass Balance Site ("MBS") [*Eicken*, 2016] (Fig. 1b, GPS coordinates in Table A1). Flat, snow-covered ice with no noticeable ridging was visible for many kilometers in all directions (Fig. 2a). Once cleared of snow (depth of 14−18 cm, -7 °C at 9 cm below the surface), the ice appeared flat and uniform. Snow was cleared prior to the collection of samples. The measured ice thickness ranged from 141−150 cm at the sampling site, which is substantially thicker than the 105 cm thickness reported at the nearby MBS. The uppermost ~10 cm (7 %) of the ice was above freeboard. At the time of our

sampling, the altimeter at the MBS had failed, so ice thickness was estimated from the thermistor string and was considered
to have large uncertainty. Ambient air temperature on the date of sampling was -9 °C at 11:00 AM. Samples collected for
analysis were subsectioned in the field at depths of 0-20 cm (top horizon), 32-52 cm (middle horizon), and the bottom 20 cm
of each core (bottom horizon).

### 2.1.2 June 2015

The second set of samples was collected on 3 June 2015 from within 30 m of the site sampled in May (Fig. 1b). The ice had
begun to form melt ponds (Fig. 2b), which we avoided during sampling. The June ice had thickness ranging from 149–159
cm, with ~21 cm above freeboard (14 %). It is likely that some of the increased ice thickness observed, compared to what
was measured in May, was due to the addition of retextured snow at the surface following a significant rain event during the
last week of May (SIZONet, 2017, observations for Utqiaġvik by Billy Adams, https://eloka-arctic.org/sizonet/), which
manifested as a layer of granular ice. Ambient air temperature on the date of sampling was -1.6 °C at 12:00 PM. Samples
collected for analysis were subsectioned in the field at depths of 0−20 cm (top horizon), 45-65 cm (middle horizon), and the
bottom 20 cm of each core (bottom horizon).

### 2.1.3 July 2015

In 2015, the landfast ice broke away from the local coastline during the third week of June (Fig. 1f). July samples were
drilled from isolated floes accessed by small boats within a radius of 32 km from Point Barrow (Fig. 1c). Floes in July varied
greatly in size, thickness, and character.
On 3 July 2015, the sea was ice-free within an ~8 km radius of Pt. Barrow; beyond this were regions of mixed ice, with both
sediment-rich and sediment-poor floes. Ice encountered near the barrier islands bounding Elson Lagoon included many
apparently grounded floes as well as some small (~7 m above freeboard), blue icebergs. Wildlife was abundant in the region,
with king and common eider, walrus, bearded seals, a grey whale, and a large pod of ~100 beluga whales observed. Cores
were drilled in three different floes: a thick (170 cm) "clean" floe (JY3-L1; with naming convention month (M, JN, JY), day
of month – location number), a small (~2 m$^2$, 86–150 cm thick in the center) sediment-rich floe (JY3-L2), and a large,
heavily-ponded floe (JY3-L3; the single core collected from this floe measured 145 cm long). At all sites, freeboard depth
was difficult to determine due to the high variability in the underside of the ice, however, roughly 10–12% of ice cored was
above freeboard. Other than the variable thickness of the floes and high sediment content in some floes, the character of the
ice was similar to the ice observed in June.
Cores collected on 10 July 2015 (JY10) came from a sediment-rich, heavily-ponded floe with an ice thickness measured in a
non-ponded part of the floe of 190 cm. Ice in non-ponded areas was solid and saline, similar to what was observed in June
and on 3 July. Cores from ponded areas of the floe (collected from ponds ~18 cm deep) were visibly highly porous (rotten)
and ranged from 58–90 cm in length. During the course of two hours of sample collection, the floe began to break up under
light wave action (winds in the region increased from ~10 to 15 knots during the course of sample collection), forcing our
team to retreat to the boat. In one case, a crack developed that connected core holes drilled across a ponded area; in another,
a large crack developed across the width of the floe.
On 11 July 2015, additional cores (64–90 cm length) were sampled from a ponded area of a clean (sediment-poor) floe of
rotten ice (JY11). As with the 10 July floe, ice in non-ponded areas was solid and saline, partially drained but not heavily
rotted. The upper portion of the ice was pitted and drained. Ice beneath melt ponds (cores collected were submerged under
8–15 cm water) was heavily rotted and drained rapidly when cored. Ambient air temperature during sampling was -1.0 – -
1.3°C.
The last cores sampled in 2015 were collected on 14 July from both ponded and non-ponded areas of two relatively thin,
clean floes (JY14-L3 & JY14-L4). Ice collected in non-ponded areas ranged from 100-139 & 80–83 cm thick and was
similar in character to the non-ponded ice of the other July floes. Ice collected in ponded areas (under 5 cm water) ranged
from 27–91 cm thick and was similar in character to the ice collected from beneath melt ponds in the other July floes.

### 2.1.4 July 2017

In summer 2017, our team returned to the offshore waters near Utqiaġvik in search of ice that had previously broken from
shore and was continuing to melt (Fig. 3). Five distinct ice floes of varying degrees of melt were sampled on 13 July 2017.
Ice thicknesses ranged from 40–110 cm. Seawater in open areas between floes had a salinity of 29.5 ppt and temperature of
+4.8 °C, as measured with a conductivity meter (YSI Model 30). Sampling on 17 July 2017 yielded ice from a single floe
with sample thickness varying between 62 and 110 cm. Pacific loons and bearded seals were observed in the vicinity.

### 2.2 Physical properties
### 2.2.1 Temperature, salinity, and density profiles

One core from each sampling site was used to measure vertical profiles of temperature, density, salinity, and pH. Ice
temperature was measured in the field immediately following core removal. The core was placed on a PVC cradle, and
temperature was measured using a field temperature probe (Traceable™ Total-Range Thermometer, Fisher Scientific;
accuracy ±1 ℃, resolution 0.1 ℃) inserted promptly into 3 mm diameter holes drilled into the center of the core at 5 cm
intervals. Horizontal pucks of the ice were then sawed at the 5 cm marks, and caliper measurements (± 0.01 cm) were taken
of two thicknesses and two diameters across the puck to estimate puck volume. Volume error values were calculated by
propagating relative variability in the thickness and diameter measurements. Pucks were then sealed in Whirlpak bags and
returned to the lab, where mass and salinity measurements were taken of melted pucks using a digital scale and conductivity
meter (YSI Model 30, accuracy ± 2 %, resolution 0.1 ppt). Bulk density was computed from the measured mass (accuracy ±
0.1 g) and estimated puck volumes.

### 2.2.2 Thin section microphotography

Representative horizontal and vertical sections were prepared from each horizon of ice for each of the three time points
sampled in 2015. Thin sections (~ 2 mm thickness) were prepared using a microtome (Leica), with the exception of some
July samples that were too fragile for microtome cutting. These fragile sections were cut as thin as possible on a chop saw (~
1 cm thickness). All cut sections were then photographed on a light table at working temperatures for each month as well as
at -15 °C. An LED epifluorescence microscope (AxioScope.A1 LED, Carl Zeiss, with EC Plan-Neofluar phase contrast
objectives) specially adapted for cold room work was used to image the thin section samples. Transmitted light
photomicrograph mosaic images were constructed from 50x magnification snapshots taken at the working temperatures for
each time point. Image software (ImageJ, Adobe Photoshop CC) and manual image analysis were used to highlight pore
spaces for pore size analysis.

### 2.2.3 X-ray micro-computed tomography

To prepare samples for x-ray micro-computed tomography ("micro-CT") imaging, 10-cm subsections of ice cores returned
from the field were stored overnight in insulated coolers in a walk-in freezer set to working temperatures. Subsections were
then placed upright in Teflon centrifuge cups (500 mL bottles with tops cut off) and spun out at -5 °C for 5 minutes at 1500
rpm to remove brine using a Thermoscientific S40R centrifuge. Samples were kept at working temperature right up to the
time they were centrifuged. The 5 minutes in the centrifuge at -5 C was assumed to be brief enough that sample
temperatures, and thus brine volume, were not significantly altered.

The masses of brine and spun-out ice were determined, and brines saved for later biological and chemical analysis. Spun-out
ice horizons were returned to the working-temperature walk-in freezer, where they were then placed upright on top of
corrugated cardboard circles placed inside the Teflon centrifuge cups. Samples were casted with dimethyl phthalate (DMP)
in an attempt to minimize structural changes during transport, storage, and processing and in order to use methods consistent
with prior micro-CT work on snow. Working temperature dimethyl phthalate (DMP) was then carefully poured down the
sides of the container in order to flood the ice samples and form casts of the brine networks in contact with the borders of the
ice core as described by *Heggli et al.*, [2009] for casting snow. The DMP was left to penetrate brine networks and slowly
freeze at the working temperature for at least 12 hours before freezing fully at -20 °C. Casted cores were then removed from
the Teflon cups, sealed in Whirlpak bags, and stored at -20 °C for later micro-CT imaging. In addition, several archived
cores from July 2015 that had been stored at -20 °C were scanned without casting to assess the effect of DMP casting on
tomography measurements.
Prepared samples were imaged at the U.S. Army Cold Regions Research Laboratory using a micro-computed tomography
high-energy x-ray scanner (SkyScan 1173, Bruker) housed in a -10°C walk-in freezer. Scans were run at 60 kV, 123 µA,
with a 200 ms exposure time and 0.6 ° rotation step. The nominal resolution was set to 142 µm pixel$^{-1}$ in a 560 x 560 pixel
field of view, which permitted fast scan times (18 minutes), resulting in low exposure of samples to excess radiation and
egregious warming (scanner chamber temperatures were recorded as ~2 °C during runs).
Shadow images generated by micro-CT were reconstructed into 2D horizontal slices using the software NRecon (Bruker).
Thermal abnormalities were corrected by performing x/y alignment with a reference scan. Samples with x/y shifts greater
than $|\Delta x| = 5$ were re-scanned. Following x/y alignment, reconstructed image histograms of linear attenuation coefficients
were clipped to 0.000 – 0.005 and the following correction factors were applied: 50 % defect pixel masking, 20 % beam-
hardening correction, smoothing level 2 using Gaussian kernel. Post-alignment shifts were determined manually and were
between -2 and 2. The ring-artifact reduction parameter was also chosen manually to minimize artifacts and was between 2
and 10 for all processed samples.
Reconstructed 8-bit 2D images were analyzed using the software CTAn (Bruker). Cylindrical subvolumes (height = 4.0 cm,
diameter = 4.97 cm) centered on the scanned sample's z-axis were selected from the original scanned samples and positioned
to capture a representative segment of the sample, avoiding sample edges. Reconstructed images were parsed into four
phases using brightness thresholding determined manually at well-defined phase local minima for each scan: air (black), ice
(dark grey), DMP (bright grey), and brine (bright). Phases were manually parsed using cutoffs based on greyscale intensity
histograms picked by a single analyst. A preliminary sensitivity analysis indicated that manual thresholding by a single
analyst was found to give more reliable results than automated thresholding methods due to relatively large variability in
brightness and contrast in reconstructed images as well as poor brightness separation between the ice and DMP phases.
Noise reduction was then applied using a despeckle of 8 voxels for ice, brine, and air, and $10^6$ voxels for DMP (a high
despeckle value ensured that only DMP-thresholded regions that connected to subvolume boundaries were included, as any
DMP "islands" are, by definition, artifacts). During the casting using DMP, air bubbles were trapped inside the solidifying
DMP. Due to the brightness order of phases (air > ice > DMP > brine) the gradient between air and DMP is incorrectly
identified as ice creating thin ice "rings" inside DMP regions of the 2D slices. This problem was resolved by using a
morphological operation called "closing", where thin (1–2 pixel) threads of ice were dilated then eroded, thus removing the
features [*Soille*, 2003]. Ultimately, DMP casting introduced artefacts in the analyzed samples, so the analyses presented in
the results of this manuscript focus only on the ice phase and the combined air + brine + DMP ("not-ice") phases.
CTAn was then used to calculate properties of the parsed phases, including 3D volume, number of 3D objects, closed and
open porosity, and anisotropy. A description of the mathematical basis for these parameters as well as detailed best-practice
methods for micro-CT imaging of sea ice can be found in *Lieb-Lappen et al.* [2017].
Further, 3D prints of the reconstructed ice-only phase were made from the micro-CT reconstructions using polylactic acid
fused deposition modeling (Flashforge Creator Pro, FDM print with Makerbot print program and layer height 0.1 mm).

## 2.3 Optical properties

Field measurements of optical properties are generally limited to estimation of apparent optical properties (AOPs), e.g.,
albedo, transmittance, and extinction. Due to the tenuous working conditions on rotten sea ice floes and instrument reliability
problems, we were not successful at obtaining estimates of *in situ* AOPs of rotten ice. Instead, we focused on assessing the
optical properties of extracted ice samples in the laboratory. Inherent optical properties (IOPs), such as scattering and
absorption coefficients and scattering phase functions, are intrinsically difficult to measure in multiple-scattering media, but
estimates from laboratory measurements can build a picture of the evolution of sea ice optical properties. In fact, estimates of
IOPs are particularly useful since they are independent of boundary conditions (e.g., ice thickness and floe size) and the
magnitude, directionality, and spectral character of the incident light field (see e.g., *Katlein et al.*, 2014; *Light et al.,* 2015).
The evolution of light scattering coefficients for sea ice as it melts determines the partitioning of solar radiation in the ice-
ocean system. *Light et al.* [2004] considered the evolution of the optical properties of sea ice samples as they warmed in a
laboratory setting, but encountered practical limitations for handling small samples of ice with large void space as the
temperature approached 0 °C. The present study specifically focused on techniques to extend our knowledge of the optical
properties of sea ice in its advanced stages of melt.
To track the evolution of how the ice in this study partitioned sunlight, a laboratory optics study was carried out. Cores for
optical property assessment were sampled alongside cores for other characterizations, returned to the lab, and stored intact at
-20 °C. The May and June cores were stored for 2–3 days prior to running the optics experiments. The July cores were
shipped back to the freezer laboratory at the University of Washington and stored for 16 months prior to optical
measurement.
To carry out optical property assessment, each core was cut into 10 cm long sections. Each section was placed in a chamber
for the measurement of light transmittance using a technique developed to infer inherent scattering properties of a sea ice
sample from a simple measurement and a corresponding model calculation (see *Light et al.*, 2015). Figure 4 shows a
schematic of this laboratory measurement, where ice samples are placed in a dark housing and illuminated from above.
Spectral light transmittance between 400 and 1000 nm wavelength of each subsample was recorded relative to the
transmittance through pure liquid water. The relative transmittance was then compared with results from numerical radiative
transport simulations using the model described by *Light et al.* [2003] for a wide range of scattering coefficients. The
scattering coefficient producing relative transmittance (at 550 nm) closest to the observed relative transmittance was then
chosen. When subsamples from a full length of ice core are measured, this technique estimates the vertical profile of the light
scattering coefficient through the depth of the ice. By directly assessing scattering coefficient, an IOP, we avoid
complications introduced by the interpretation of AOPs (e.g., albedo, total transmittance measured *in situ*), notably
differences in total ice thickness and incident solar radiation conditions (e.g., diffuse/direct), as well as other physical
boundary conditions. In each case, samples taken from ice sitting below freeboard were placed into the sample chamber and
then gently flooded with a sodium chloride and water mixture in freezing equilibrium (temperature and salinity) with the
sample. Light transmission was measured while the sample was flooded. Sample measurement was fast, with each sample in
the chamber for less than one minute. It is probable that the liquid did not completely fill the pore structure of the ice
samples, however, the visible appearance of the samples indicated a dramatic reduction in backscatter during the flooding
process, suggesting that flooding was effective.
Samples were run in two modes. In the first mode, samples were analyzed promptly after removal from the ice. These
samples represent snapshots of the rotting process as it occurs naturally. The second mode was run in attempt to use light
scattering measurements to inform our understanding of ice rotting processes. To do this, an archived May core was cut into
10 cm thick sections and placed in an insulated box in the freezer laboratory. The sections were stored standing upright and
were placed on a wire rack such that the melt water drained away from the remaining sample material. Initially, the freezer
temperature was set to -8 °C, but once the experiment commenced, the temperature was increased gradually every 24 hours.
The sample density and vertical scattering profile were measured at each temperature step (-6, -5, -4, -3, -2 °C over a one
week period.) This attempt to artificially rot the ice was documented using the optical measurements with the hope that such
a measurement would inform our efforts to simulate rotten ice in the laboratory.

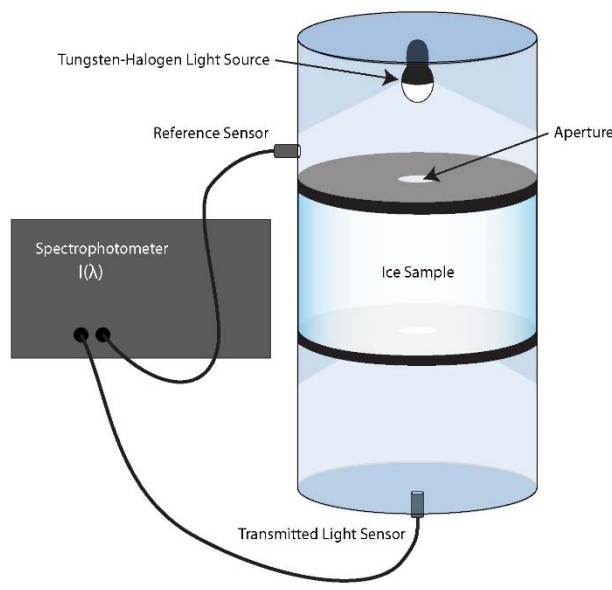


Figure 4. Schematic depicting laboratory setup for measuring light transmittance through 10 cm tall ice core samples. Adapted from *Light et al.*, [2015].

## 3 Results

### 3.1 Ice core samples

Figure 5 shows photomosaics of the microstructure in representative cores collected at the different time points and from different rotten floes. The series shows the progression from recognizable congelation ice in May, to the development of a retextured snow layer in June, to the chaotic appearance of the ice structure in July.

#### 3.1.1 May

In May, the interior of the ice was relatively translucent due to the small, isolated nature of brine and gas inclusions, a result of the still relatively low temperature of the ice. Obvious brighter white bands of concentrated bubbles were present within the ice. A weak layer was present in several cores between roughly 32–45 cm, which defined breaking points of the corresponding middle horizon samples. May cores also exhibited a brown discoloration in the ice proximal to the ice-ocean interface, which is indicative of algae; microscopy confirmed the presence of abundant pennate diatoms in ice bottom samples.

#### 3.1.2 June

In June, the ice interior did not appear visibly distinct from May ice except for the upper surface of the ice. Significant rains during the last week of May fell on the snow-covered ice, saturated the snow, and refroze. This produced a retextured snow layer that occupied the upper 20 cm of the ice and was composed of grainy, bright ice with low structural integrity. Ice below the retextured snow layer was soft and saline. Telltale discoloration in the bottom ~2 cm of sampled cores, albeit fainter than May, indicates the algal cells had not yet completely sloughed from the ice. Additionally, green coloration of collected seawater indicated the presence of an algal bloom in the water column (7 ppt, 0.0 °C) at the base of the ice.

#### 3.1.3 July

Ice collected in July 2015 and July 2017 was highly variable. Cores collected on 3 July 2015 were largely similar in character to samples collected in June, including an apparent retextured snow layer in the upper ~10 cm of the ice. The ice-bottom algal discoloration present in the May and June ice, however, was absent in July. Seawater had no apparent algal bloom, and measured temperatures of 1.5 – 4 °C between floes and 0.2 °C directly below several sampled floes.

Ice sampled in mid-July in both 2015 and 2017 was found to be in various stages of rot. Ice sampled from thick floes was similar in character to the June 2015 and 3 July 2015 ice in non-ponded regions, but distinctly rotten below melt ponds.

Uniformly thin floes were rotten throughout in both ponded and non-ponded regions. Visually, rotten ice was devoid of the
microstructural inclusions that characterized the May and June ice interior, instead appearing to have large, isolated pores,
and a more chaotic structure. When cored, rotten ice crumbled or broke at many points along the length of cores, rendering it
difficult to handle. Rotten ice drained copiously when cores were removed from drill holes, and the bottom portion of rotten
cores consisted of optically clear, fresh ice drained of brine and characterized by large (cm-scale) voids.  Figure 6 shows
photomosaics of cores sampled on 14 July 2015 at Location 3. Images show variations in ice texture depending on whether
the ice was ponded or non-ponded, although both types do appear to have at least some scattering layer with bright white
appearance.  Many cores had holes exiting the bottom of the ice that were large enough to stick a finger into, although we
did not have a means to quantitatively assess how vertically extensive these drainage tubes were.

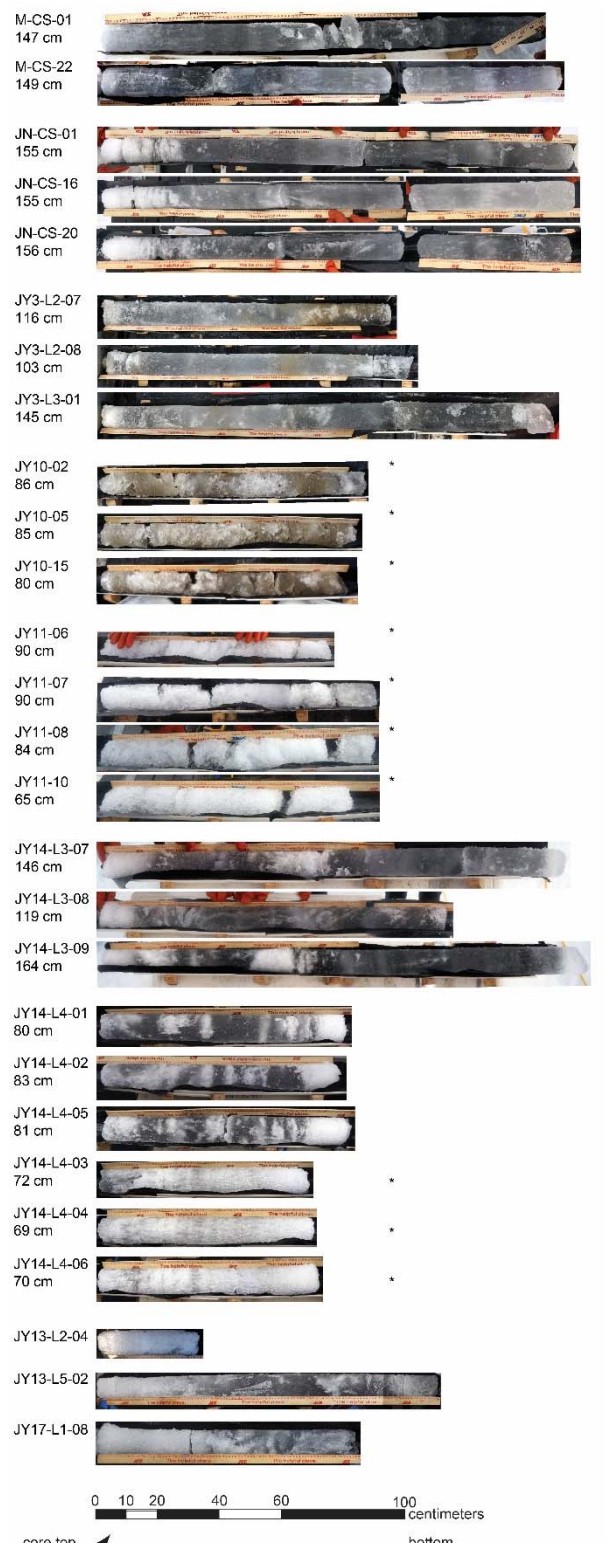

Figure 5. Photomosaics of representative cores collected and analyzed in this study showing the sequence of rot. Core names correspond to samples discussed elsewhere in this paper and are coded by sample site (as shown in Figure 1). The measured ice thickness at each core hole is indicated. For the JY14 samples, measured core length is indicated instead of ice thickness. Due to variability in the ice bottom, spreading or compression of weak layers, and artefacts of image stitching, core images, which are shown to scale, may not match the measured ice thickness. Asterisks (*) indicate cores collected from submerged ice. Note the brown algal bloom layer visible in the bottom of the May core and faintly visible at the bottom of the June core, and the bright layer of retextured snow at the top of the June cores.


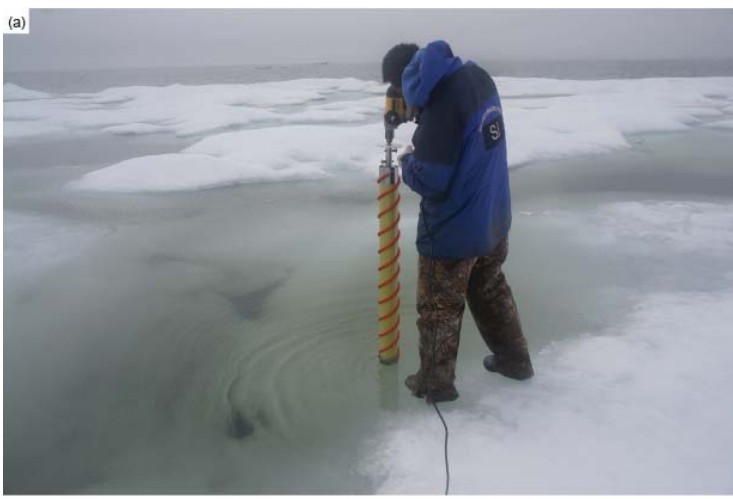

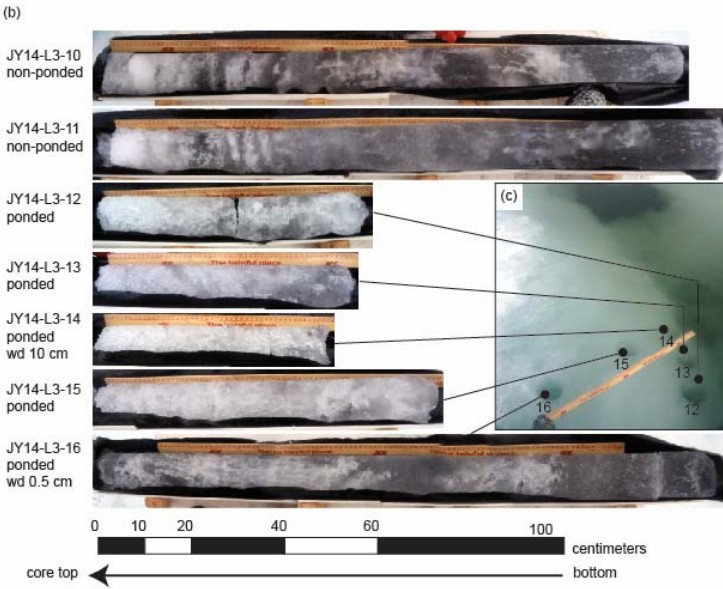


Figure 6. (a) Photograph of the part of a rotten floe sampled on 14 July 2015 (JY14-L3). Ponded and non-ponded areas visible in the picture were drilled for core samples. (b) Several cores drilled from JY14-L3 floe (JY14-L3-10–JY14-L3-16) showing the variability in length and character of ice from non-ponded areas vs. from beneath pelt ponds. (c) Region of ponded ice that was drilled to collect the ponded ice cores shown in (b). Cores JY14-L3-12–JY14-L3-14 were drilled from ice at a water depth ("wd" in legend) of ~10 cm, JY14-L3-15 was drilled from intermediate depth, and JY14-L3-16 was drilled from a water depth of ~0.5 cm.


**3.2 Physical properties**

**3.2.1 Temperature, salinity, and density profiles**

The May temperature profile had values as low as -8 °C at the snow-ice interface; below that, temperatures increased with
depth (Fig. 7). By June, the entire depth of the ice had warmed above -1 °C, with the lowest temperatures measured in the
middle sections of cores. By July, the ice was approximately isothermal, with temperature 0 °C. These profiles generally
agree with observations at the MBS and are typical of other investigations in the area (e.g., *Zhou et al.*, 2013).
Bulk salinity profiles (Fig. 7b) were also consistent with prior published observations. May ice showed the classical C-
shaped salinity profile with enhanced salt content near the upper and lower boundaries (10 ppt) and lower salt content (< 5
ppt) in the interior of the ice. By June, significant fresh water flushing from rain and snow melt reduced the salt content in
the upper portions of the ice. The July profiles showed evidence of prolonged flushing, with salt content approaching zero in
some cores.
Density values (Fig. 7c) measured in this study in May and June averaged 0.91 and 0.87 g cm$^{-3}$, respectively, with the lowest
density values (as low as 0.63 g cm$^{-3}$) found in the uppermost portions of the June core. Relative measurement errors
(resulting from variability in multiple measurements of height and diameter from each puck, calculated by propagating errors
in the density calculation) calculated for May and June samples were typically <6 % in May and <10 % in June except for a
few outliers, while July samples had many samples with measurement errors >10 % because of difficulty determining a
volume for the irregular sample shapes.

**3.2.2 Thin section microphotography**

Thin sections show the evolution of the ice structure as it warmed (Fig. 8). Each of the microphotographs in Fig. 8 is a
stitched composite of 20 individual images taken at 25x magnification. The May and June images clearly show individual
brine and gas inclusions.
Inclusions in the May sample had average size of 80 μm (9−577 μm, standard deviation 62 μm, 162 inclusions resolved)
with an inclusion number density of 32 mm$^{-3}$. The average size of inclusions in the June sample increased to 221 μm
(61−587 μm, standard deviation 105 μm, 103 inclusions resolved) while the number density decreased to 19 mm$^{-3}$. The July
sample exhibited notably larger and fewer inclusions. Due to the difficulty in preparing thin sections from fragile, rotten ice
and the large size of pores, only nine individual inclusions were completely resolved from July samples. Despite poor
statistics, the average size of the inclusions in the July sample was 3 mm (range 1−5 mm) and the estimated inclusion
number density was 0.01 mm$^{-3}$. The reduced number of resolved inclusions with time is expected as the inclusions enlarge
(due to freezing equilibrium) and merge.

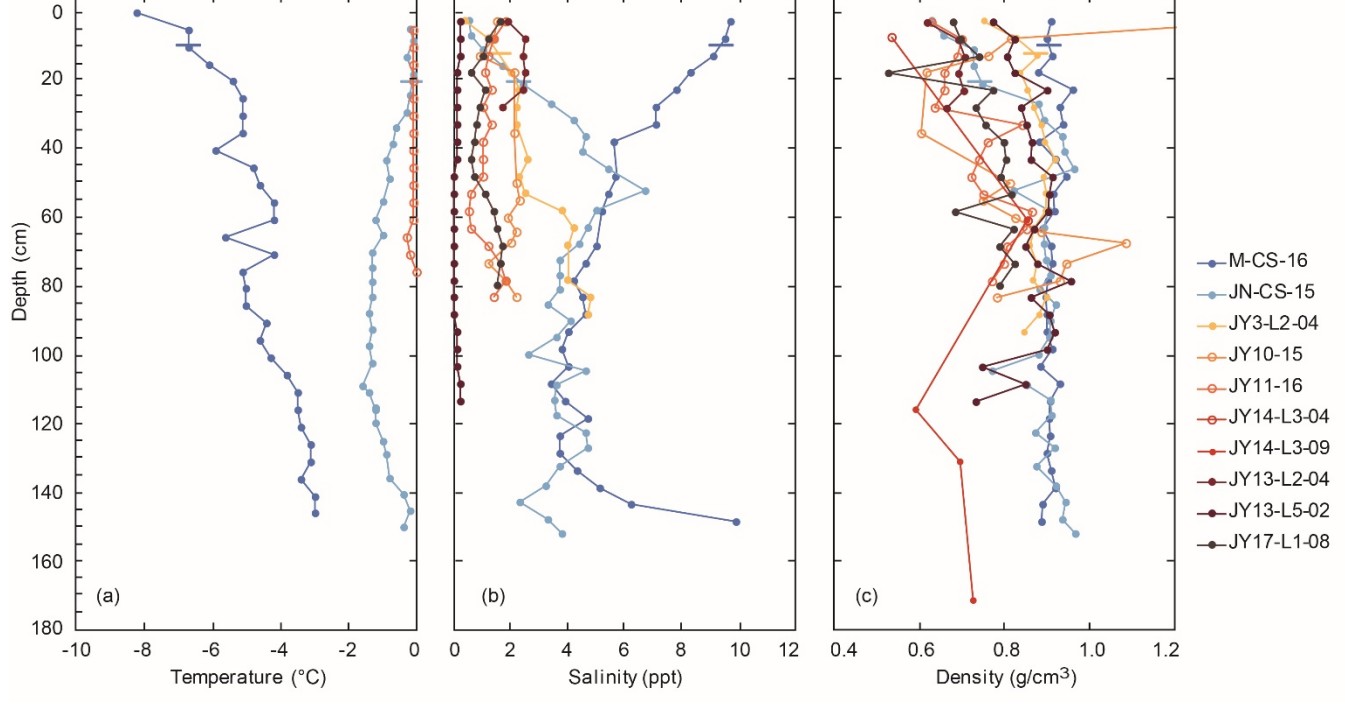


Figure 7. Ice core profiles of (a) temperature, (b) salinity, and (c) density showing changes in ice properties over the course
of summer melt. Open circles indicate cores of ponded ice. The position of freeboard is indicated by a horizontal bar in cores
where freeboard was measured; note that for ponded ice, the ice was below freeboard. Ice cores analyzed as follows: May
(M-CS-16, dark blue), June (JN-CS-15, light blue), July 2015 (3 July core, JY3-L2-04, yellow; 10 July rotten core from
sediment-rich, ponded ice,  JY10-15, orange; 11 July rotten core from ponded ice,  JY11-16, dark orange; 14 July rotten core
from ponded ice, JY14-L3-04, red, open circle; 14 July rotten core from non-ponded ice, JY14-L3-09, red, closed circle),
and July 2017 (13 July Floe 2 JY13-L2-04, magenta; 13 July Floe 5 JY13-L5-02, purple; 17 July Floe JY17-L1-08, brown).

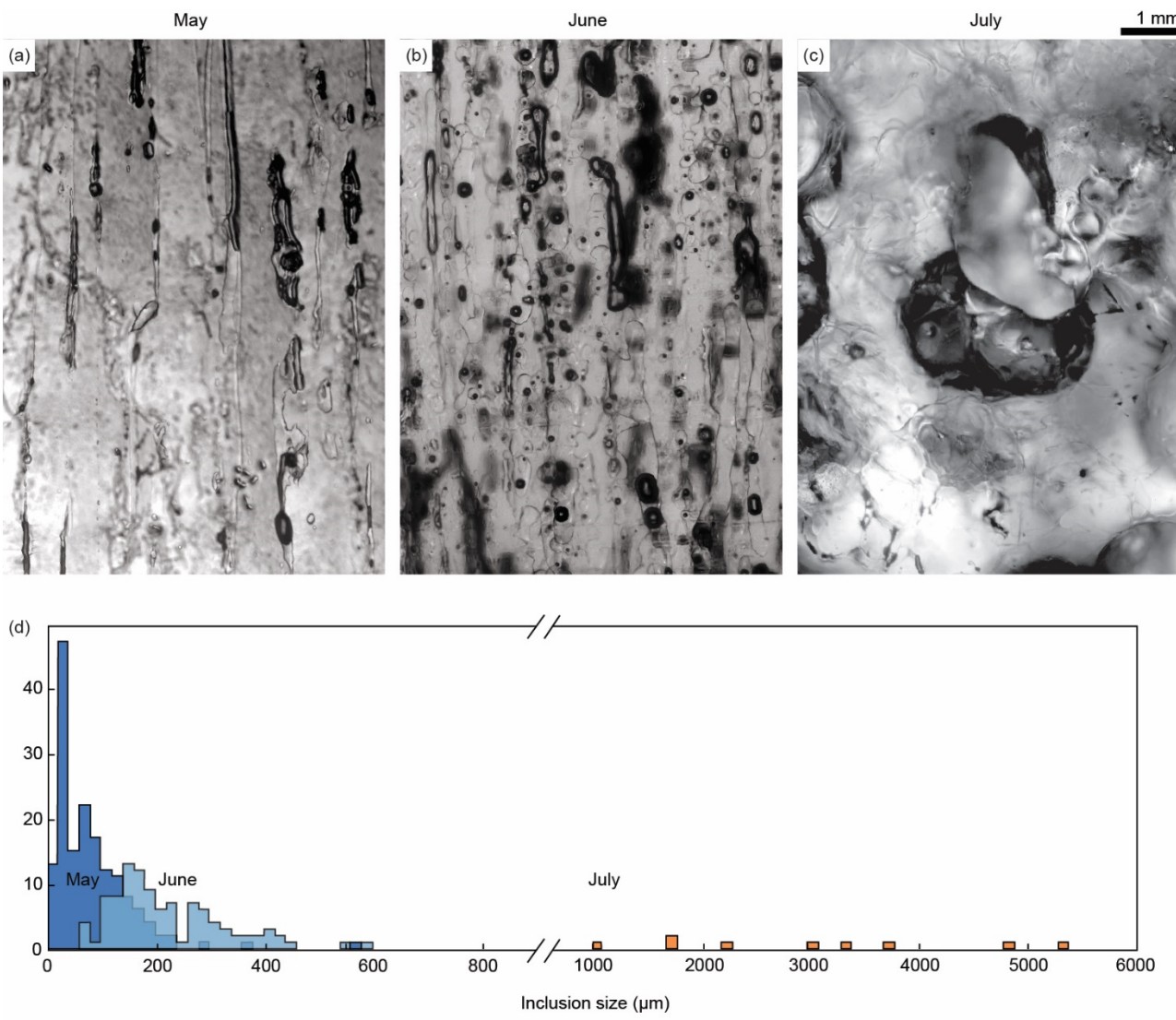


Figure 8. Ice sample vertical thin section transmitted light photomicrograph mosaics (a–c) and histogram of measured pore sizes (d) from May, June, and July ice samples, shown at the same scale. (a) Vertical thin section from middle horizon of M-CS-8 core (depth = 32–52 cm) showing vertical brine channels. (b) Vertical thin section from middle horizon of JN-CS-22 core (depth = 45–65 cm) showing enlarged brine channels. (c) Vertical thin section from middle horizon of JY10-CS-11 (depth = 32–42 cm) showing pore space. (d) Pore size histogram indicating the maximum dimension of pores measured from thin sections collected in May (dark blue), June (light blue), and July (orange).


### 3.2.3 X-ray micro-computed tomography

Calculations done on 3D reconstructions generated from micro-CT show a significant evolution in the internal structure of
ice during the course of melt and help define "rotten" ice. Figure 9 shows reconstructions of the ice-only phase (top row),
reconstruction of the not-ice phase (air + brine + DMP) with objects of different sizes color coded as blue ($<0.11$ cm$^3$), green
($0.11$–$1.15$ cm$^3$), and red ($>1.15$ cm$^3$) showing the evolution toward larger pores and channels in rotting ice (middle row).
Note that micro-CT analyses only resolve structures with a short dimension $> 284$ μm, (derived from the 8 voxel despeckle
that was applied) which is significantly larger than the average inclusion size observed in the microscope imagery for both
May and June. The bottom row shows monochrome photographs of 3D prints made from the four reconstructions.
Porosity is defined as the percentage of total volume occupied by pores, as measured from the ice-only phase perspective
such that the porous space is derived from air, brine, and DMP.  Porosity in DMP-casted May and June horizons (excluding
June top horizons determined to represent a retextured snow layer) ranged from 0.5–7.5 % by volume (Fig. 10a). In contrast,
the DMP-casted rotten core (JY11-06) had a range in porosity of 37.5–47.9 %. For non-casted rotten cores measured, the
porosity ranged from 7.6–23.1% (mean = 15.5 %) in a sample collected from below a melt pond (JY11-19), and from 5.7–
46.0 % (mean = 21.6 %) in samples collected from bare, non-ponded ice (JY13-2 and JY13-4). Bare ice had the highest
porosity values in the upper 10 cm (24.7–46.0 %), corresponding with a retextured snow layer. Similarly, two sample
volumes selected from retextured snow layers of June ice exhibited extremely high porosity values of 48.9 % and 53.6 %.
In addition to becoming generally more porous, the nature of pores in the ice changed as melt progressed (Fig. 10b). Open
pores were those pores connected to the exterior surface of the volume analyzed, while closed pores were those fully interior
within the 77.6 cm$^3$ volumes analyzed. In May, closed pores comprised 26–72 % of the total pore volume (mean = 51%). In
June, the percent by volume of closed pores was similar (mean = 42 %) except for the uppermost retextured snow layer (0–3
% closed pores by volume). In July, this was markedly changed: >74 % of pore volume in all samples (casted and uncasted)
of July ice was open, i.e., in communication with the surrounding ice. Most July samples were >98 % open pore space by
volume (mean = 96 %, median = 99 %); samples with <90 % open pore space were from the interior of the JY14 samples. In
addition, the number of closed pores in the normalized unit volume decreases from May and June to July (Fig. 10c). The
May cores and June middle horizons have the highest closed pore densities (31–94 cm$^{-3}$ with mean = 61 cm$^{-3}$, and 2–63 cm$^{-3}$
with mean = 26 cm$^{-3}$ in May cores and 44–63 cm$^{-3}$ measured in June middle horizons). In June, the density of closed pores in
the top and bottom (8–11 cm$^{-3}$, and 2–10 cm$^{-3}$, respectively) decrease, creating a reverse C-shaped profile. In July, the
density of closed pores is uniformly low throughout the cores measured (16–36 cm$^{-3}$ with mean = 24 cm$^{-3}$, and 1–16 cm$^{-3}$
with mean = 6 cm$^{-3}$ in casted and non-casted July cores, respectively). Both metrics indicate that connected (open) pores
dominate in July. This follows from larger pore sizes, as quantified by the 2D structure thickness metric, which measures the
mean maximum diameter of 3D objects. In May and June, pores averaged <5 mm along their longest axis (1.7–3.3 mm,
mean = 2.4 mm, again with the exception of the June retextured surface snow and a 32 mm outlier value in one June middle
horizon). In rotten July cores, pores enlarge substantially (4.2–17.0 mm, mean = 8.1 mm). The trend toward more connected
pores is most pronounced in the upper- and lowermost layers of the core.

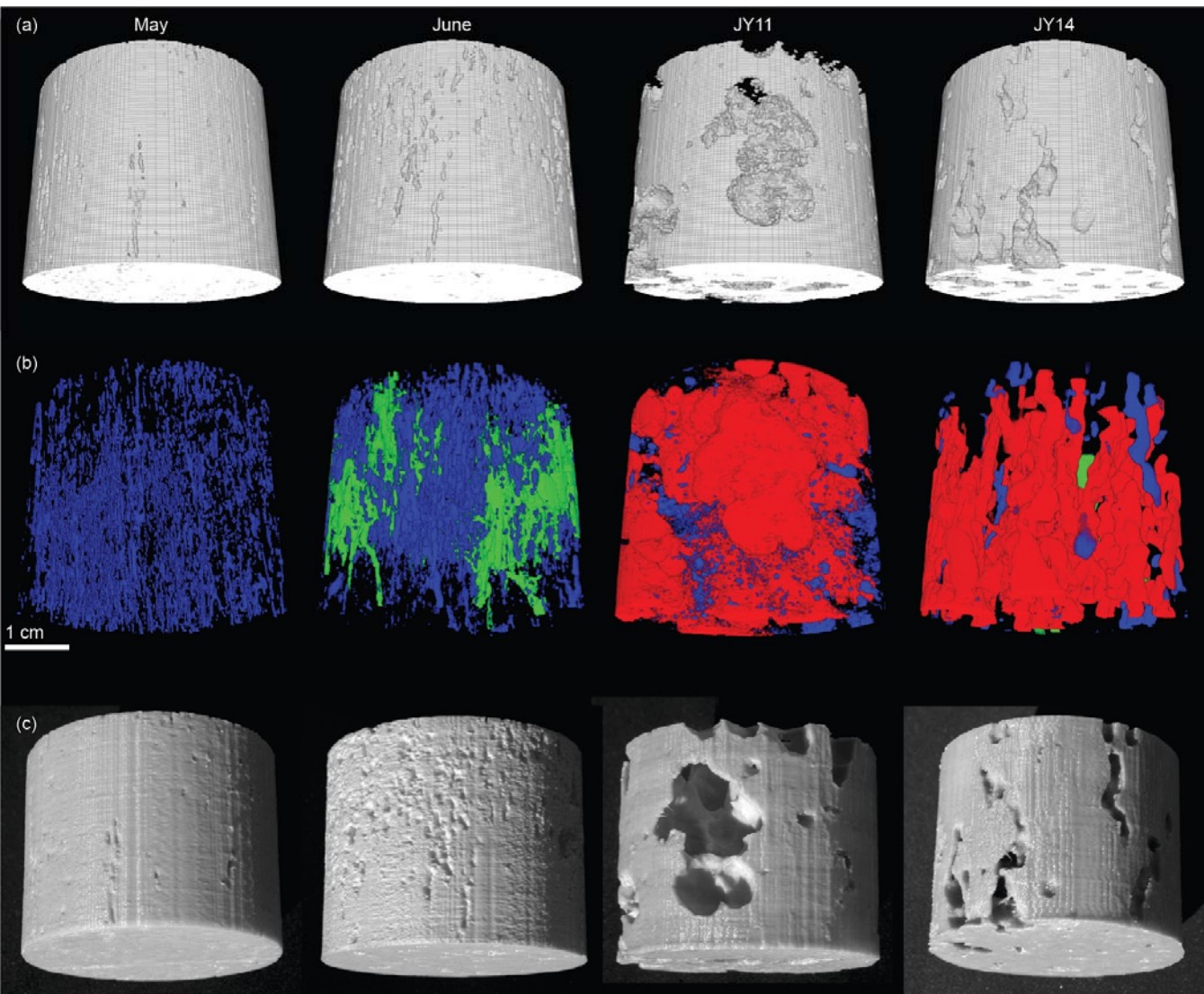

Figure 9. 3D reconstructions from micro-CT scans of middle horizon cuts of cores collected in May, June, and July (JY11
and JY14) 2015 showing the evolution of pore space. Series (a) shows the reconstruction of the ice-only phase. Series (b)
shows the reconstruction of the not-ice phase (air + brine + DMP) with objects of different sizes color coded as blue (<0.11
$cm^3$), green (0.11–1.15 $cm^3$), and red (>1.15 $cm^3$) showing the evolution toward larger pores and channels in rotten ice.
Series (c) shows monochrome photographs of 3D prints of the reconstructed ice-only phase.

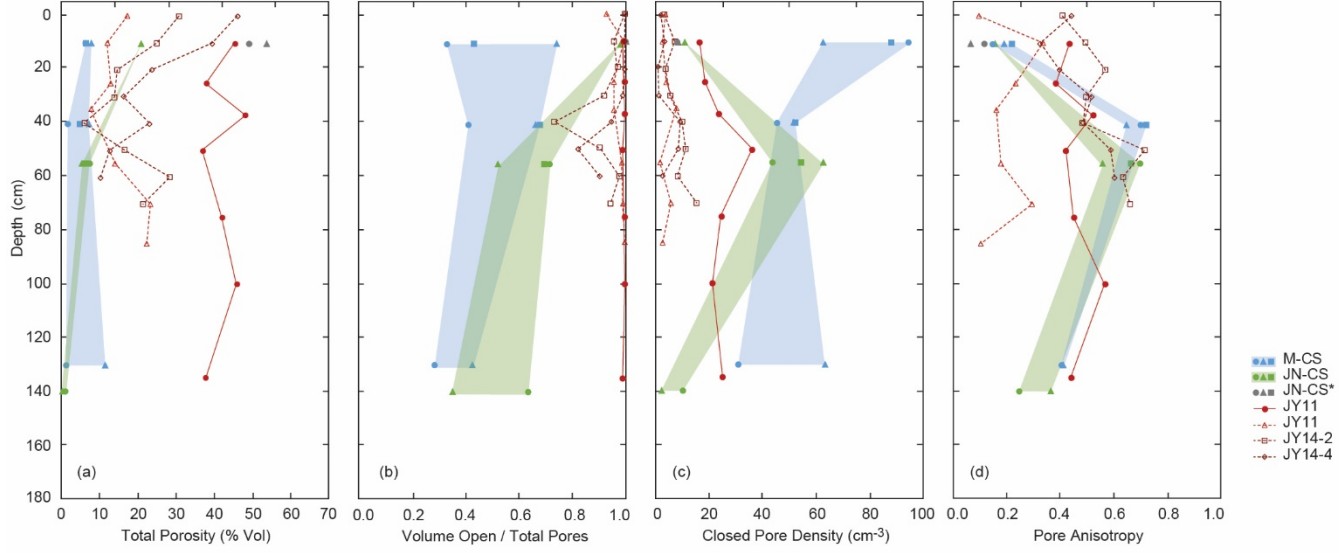

Figure 10. Sea ice internal pore properties calculated from 3D reconstructions of micro-CT scans of cuts of cores collected in May, June, and July. (a) Total porosity as percent of the analyzed volume. (b) Volume of open pores vs. total volume of pores in the analyzed volume. (c) Average spatial density of closed pores in the analyzed volume. (d) Anisotropy of pores in the analyzed volume. Depths indicate the in situ depth within the ice of the top of the volume of interest used for the calculations. Colors indicate sampling month: May (M-CS; blue), June (JN-CS; green), and July (JY11 and JY14; red and dark red, respectively). Shaded blue and green fields represent the range of values measured in replicate ice horizon samples in May and June, respectively. Closed markers indicate DMP-casted samples; open markers indicate un-casted samples. Values calculated from a volume believed to be representative of a retextured snow layer in the uppermost June samples are represented by grey symbols. For all points, the volume analyzed was a 77.6 cm³ cylinder (diameter = 4.97 cm, height = 4.0 cm) selected from a representative portion of the interior of the cut sea ice horizon.

Anisotropy roughly indicates deviation from spherical structures, with a value of 0 indicating a perfectly isotropic sample (identical in all directions) and 1 indicating a perfectly anisotropic sample (fully columnar). This definition for degree of anisotropy ("DA") follows from the equation DA = 1 – [minor axis / major axis] (*Odgaard*, 1997). In sea ice, the highest degree of anisotropy corresponds to elongated brine channels in columnar ice [*Lieb-Lappen et al.*, 2017]. Anisotropy (Fig. 10d) in the not-ice fraction (air + brine + DMP) of May and June samples followed a reverse C-shaped profile (cf. *Lieb-Lappen et al.*, 2017), with the highest degree of anisotropy found in middle horizons (0.43−0.72) and lower anisotropy in the top and bottom horizons (0.14−0.41). While this may seem counterintuitive, a simple analogy using pasta may be helpful. Pasta shells (rounded) would be a good way to visualize an isotropic assembly of pores. Spaghetti (pre-cooked) is clearly anisotropic. However, the strongest anisotropy could be represented by pre-cooked spaghetti still in the box. If the uncooked

spaghetti were spilled on the floor, it would become more isotropic, even though each individual piece is anisotropic.
Spaghetti in the box is a good analogy for the pore spaces in the mid-horizon. Horizontal connectivity in the bottom horizon
makes that pore space less anisotropic.
In the rotten July cores, the C-shaped profile disappeared entirely. In the JY11 sample analyzed (from ponded ice), the
middle portion of the core became more isotropic (0.38–0.57 in the DMP-casted sample, 0.24–34 in the uncasted sample),
indicating a rounding of the core center brine channels. This trend was not apparent in the JY14 (thinner rotten floe of non-
ponded ice) sample, however, in all July cores analyzed, the upper layer had a generally greater anisotropy value than core
middle values, perhaps indicative of vertical channel formation in the upper portion of the ice due to melt and draining from
the upper portion of the ice.

### 3.3 Optical properties

As the sea ice cover progresses through the onset and duration of melt season, its optical properties respond to increased
temperature and the absorption of increasing amounts of solar radiation. Typically, the albedo of the ice cover decreases (less
light backscattered to the atmosphere) and its transmittance increases (more light propagating into the ocean). The bulk of
this effect, however, is due to the loss of accumulated snow and the widespread formation of melt puddles on the ice surface
[*Perovich et al.*, 2002]. While this net effect dominates the surface radiation balance, it overlooks effects due to changes in
the properties of the ice itself. As the ice warms and becomes porous, permeable, and rotten, increases in void space increase
the total amount of internal ice / liquid and ice / air boundary, and would thus be expected to increase total scattering.
Increases in ice scattering should promote higher albedo and lower transmittance—exactly opposite the behavior of the
aggregate ice cover.
The results of the laboratory optical measurements are shown in Fig. 11. Vertically resolved profiles of scattering coefficient
are shown for ice obtained in April, May, June, and July. The April ice was extracted in the same vicinity as the May and
June samples during an unrelated field campaign in 2012. In addition to the temporal trend of sampled ice, optical property
assessment was also carried out for a May sample subjected to controlled melt in the laboratory (open circles). Scattering
coefficients generally increased with time and individual profiles were typified by the characteristic c-shape (higher
scattering at top and bottom of the column, lower scattering in the middle) also seen in typical salinity profiles.

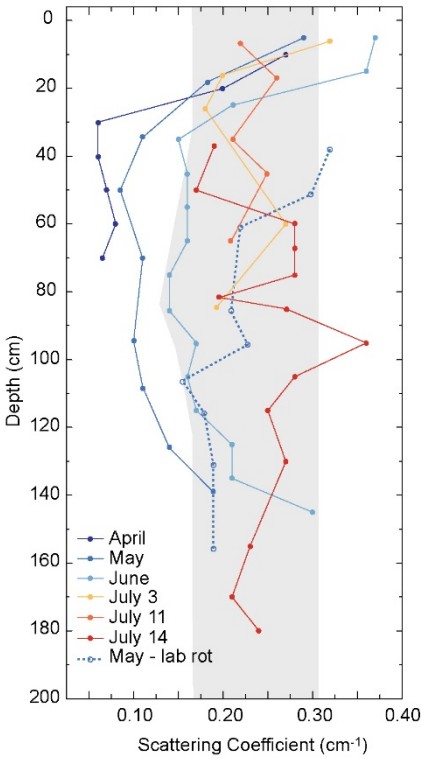


Figure 11. Vertically resolved scattering coefficients of sea ice measured during each phase of the field campaign. Coefficients are inferred from laboratory optical transmittance measurements (after *Light et al*., 2015) and interpretation of a radiative transport model in cylindrical domain (*Light et al.,* 2003). April profile included to show spring ice was measured on ice sampled in 2012 at a comparable geographic location. The May lab rot profile is for ice extracted in May during field campaign, and then warmed in the lab prior to sub-sample preparation. Shaded area shows the range of measurements on melting multiyear ice (*Light et al*., 2008) and melting first-year ice (*Light et al*., 2015).


## 4 Discussion

### 4.1 Physical characteristics of rotten ice

As sea ice warms, its microstructure changes as inclusions of brine and gas enlarge as required to maintain freezing
equilibrium. This has been well established theoretically [*Cox and Weeks*, 1983], as well as in laboratory experiments
[*Perovich and Gow*, 1996; *Light et al.*, 2003] for ice with isolated inclusions of brine and gas. This study addresses the
limits of sea ice microstructure when natural ice is in advanced stages of melt, where these inclusions are typically no longer
isolated, but rather are in connection with the ocean and/or the atmosphere.
The equations of *Cox and Weeks* [1983] describe the phase relations of sea ice for temperatures less than or equal to -2 ℃
and where the bulk ice density describes a volume containing liquid brine and gas−both in equilibrium (freezing equilibrium
with the ice in the case of brine and phase equilibrium with the brine in the case of gas). *Lepparanta and Manninen* [1988]
expanded this treatment to include temperatures above -2 ℃. In the case of ice in advanced melt, the ice temperature would
be expected to be always close to 0 ℃. Furthermore, most sample volumes will typically include void spaces that are in
connection with the atmosphere or ocean and hence may not conform to the requirements of freezing or phase equilibrium
(e.g., brine inclusion size will not necessarily shrink if the temperature decreases). As a result, expected changes in the
microstructure−and ultimately, the mechanical behavior−of sea ice at most times of the year should not be expected to
pertain to changes experienced during late summer.
Photos of ice core samples shown in Fig. 5 illustrate the evolution of the ice structure. Early in the season, the majority of the
interior ice (areas away from the top and bottom) appears mostly translucent and often milky with the exception of isolated
bright, bubble-rich weak layers. As the season progresses, more of the ice appears opaque, losing its transparency (Fig. 5).
This highly scattering ice results from merging, connecting, and draining inclusions. This effect is clearly seen in the cores
that were submerged when extracted (e.g., the cores indicated with * in Fig. 5, and cores shown in Fig. 6), but can also be
seen in the JY13-L1 and JY13-L2 cores, which were not submerged when sampled.
Submerged cores appear to have more porous ice structure. We hypothesize this is due to additional heating of submerged
ice. This heating may come as a result of increased absorption of radiation as swamped or ponded ice will not maintain a
substantial surface scattering layer, and as a result, its albedo is typically lower [*Light et al.*, 2015], and more sunlight is
absorbed within its interior. Or it may result simply from the contact between this ice and sunlight-warmed water. It is also
possible this additional melting serves to enhance the connectivity of this ice to the ocean, promoting the invasion of
seawater−and any associated heat−from beneath.

### 4.1.1 Temperature, salinity, and density profiles

Rotten ice is isothermal, having warmed to approximately the freezing temperature (0 ℃) of fresh water. Correspondingly,
core samples of rotten ice extracted from the ocean typically drain any associated liquid rapidly. Accordingly, this ice is
much fresher than earlier-season ice, with salinity values < 3 ppt through most of the core, indicative of a loss of much of the
brine that characterizes earlier-season ice (see Fig. 7b). The May salinity measurements show the classic 'c-shaped' salinity
profile indicative of first-year ice yet to experience summer melt. By June, the salinity profile shows freshening at the ice
bottom, likely associated with the onset of bottom ablation. It is  also possible that this freshening resulted from increased
brine drainage during core sampling of ice with enlarged pore space. However, the optical transparency of the bottom
portion of the ice when sampled as well as the micro-CT data imply that little closed porosity remains in rotten ice—the ice
is snaked through with large drainage tubes. Additionally, the top of the June ice shows significant freshening. In this
particular year at this location, this change is likely related to the presence of retextured snow at the ice surface, which would
be expected to be very fresh. It may also result, in part, from the onset of surface ablation and the ensuing fresh water
flushing that would be expected this time of year. The July ice was almost completely devoid of salt. This is expected, due to
the prevalence of a connected pore structure and the significant flushing and drainage of virtually all salt in the ice.
Density profiles (Fig. 7c) reflect changes in temperature, bulk salinity, and structure. We observed a marked decrease in
density corresponding to summer melt, a result of the dramatic increase in porosity that defines rotten ice. May and June
profiles had density measurements centered around 0.9 g cm$^{-3}$ and showed little variability except for reduced density in the
upper portions of the June core, likely resulting from the prevalence of the observed retextured snow. July profiles had even
further reduced density, with values reaching as low as 0.6 g cm$^{-3}$, reflecting void spaces in the ice following the rapid
draining of seawater from the ice, and was much more variable. For comparison, the density of core horizons (measured
using the same technique) taken in melting Arctic pack ice in July 2011 had similar values between 0.625–0.909 g cm$^{-3}$
[*Light et al.*, 2015]. Normally, sea ice with significantly smaller bulk density would be expected to float higher in the water
and thus have larger freeboard. But the density reductions that occur during advanced melt result from large void spaces
within the ice that are typically in connection with the ocean. As a result, such ice can have small freeboard, even if total ice
thickness is still relatively large.
It is worth noting that sediment loading did not appear to influence the density and structure of rotten ice. Rotten cores
collected on 10 July 2015 came from a floe with a visibly high sediment load, while rotten cores collected on 11 July 2015
and in July 2017 had much less sediment (Fig. 5). For all July cores, measured density values were similar within the large
range of measurement error. Salinity in the core collected from a sediment-rich floe was, however, somewhat higher than the
cores collected from "clean" floes.

### 4.1.2 Internal structure: porosity, connectivity and implications of rot

The number and size of brine inclusions identified in this study through the microscope imagery is commensurate with the
number and size of inclusions documented by *Light et al.* [2003]. That study reported a brine inclusion number density range
of 24 mm$^{-3}$ to 50 mm$^{-3}$ from ice sampled in May, offshore from Utqiaġvik in a similar vicinity as the present study. The
number densities observed in May ice in the present study were 32 mm$^{-3}$ in May, well within the range identified by the
earlier study. The earlier study showed brine inclusion number densities to decrease with increasing temperature, up to a
point, but did not follow the ice into advanced melt. The present study documents decreases in inclusion number density
from 32 mm$^{-3}$ in May to 19 mm$^{-3}$ in June to 0.01 mm$^{-3}$ in July. While these values are consistent with the earlier findings,
they also extend the results much further into melt than has been previously attempted. In particular, the micro-CT work is
useful for sampling much larger sample volumes, and thus central for estimating size and number distributions for the July
ice.
Porosity (Fig. 10a) is low in May, with values less than 10 %, and increases as the ice warms and melts. By July, the micro-
CT-determined porosity approached 50 %, commensurate with densities measured as low as 0.6 g cm$^{-3}$ and our general
observations that this ice was highly porous, containing obvious channel structures with that were clearly connected. There
were differences in the handling of cores used for direct density measurement and cores used for micro-CT imaging. In
particular, cores used for density measurement were extracted from the ice immediately prior to measuring their dimensions.
In contrast, samples taken for micro-CT imaging spent several hours transiting to the laboratory, which may have enhanced
brine loss and structural change. In addition, samples casted for micro-CT imaging were centrifuged prior to casting. It
would thus be expected that the micro-CT-derived porosity measurements could yield estimates with less included fluid than
the density measurements made closer to in situ conditions. Similarly executed micro-CT measurements have quantified
included air volumes in growing winter sea ice [*Crabeck et al.*, 2016], where the gas phase was clearly distinguished from
the brine phase, but the total pore space did not increase above 11 %, which is far smaller than the ultimate pore space
observed in this study.
The permeability, and hence pore structure, is central to the hydrological evolution of summer sea ice [*Eicken et al.*, 2002].
This suggests that the documentation of highly permeable ice with large porosity may be central to understanding the mass
balance of modern ice covers late in the summer melt season. In particular, Eicken et al. [2002] outlined a mechanism for
significant ice melt whereby warmed surface waters run off the ice and accumulate beneath areas with shallow draft late in
summer, and this pool of warmed fresh water experiences convective overturn and is entrained within the open structure of
melting ice. It is expected that further melting from this additional heat could exacerbate the decay and structural frailty of
the melting ice, literally melting it from the inside out.
The pore anisotropy results shown in Fig. 10d reinforce the overall trend that as the season progresses, the ice structure
homogenizes, losing its characteristic c-shape. Where strong vertical gradients in anisotropy existed in May and June, the
July ice is more uniform. Our findings are consistent with those of *Jones et al.* [2012], which used cross-borehole DC
resistivity tomography to observe increasing anisotropy of brine structure as early spring (April) ice transitioned to early
summer (June) ice. In that work, the brine phase was found to be connected both vertically and horizontally and the
dimensions of vertically oriented brine channels gradually increased as the ice warmed.
There remain notable limitations associated with the characterization of sea ice using micro-CT techniques. Many small
brine inclusions were not counted owing to the limited spatial resolution of the technique. Furthermore, the casting technique
that was employed appears to have introduced artifacts, especially in connectivity. From all the derived properties (porosity,
connectivity, and anisotropy), it appears that the introduction of the casting media may have forced channel connections
where perhaps they did not exist naturally. However, the trend in casted samples and the values measured for uncasted
samples reflect the substantial changes in ice character that are apparent in the field.

**4.2 Optical evolution of rotting ice**

Increases in effective light scattering coefficient over the course of seasonal warming are shown to be approximately 5-fold
for the interior ice studied here (Fig. 11). The overall trend of increasing scattering with time as the melt progresses is a
result of the connecting and draining microstructure, as assessed in the microstructure and tomography analyses. Relative
increases in the scattering would be expected to scale by the inclusion number density multiplied by the square of the
effective inclusion radius (see *Light et al.*, 2003). Observed mean inclusion sizes increased from average May size of 80 μm
to average June size of 221 μm to average July size of 3 mm. Observed number densities decreased from 32 mm$^{-3}$ (May) to
19 mm$^{-3}$ (June) to 0.01 mm$^{-3}$ (July). These changes correspond to relative scattering coefficient magnitude changes of 1: 4.5:
0.4, which would predict a scattering coefficient increase from May to June by a factor of 4.5, and a decrease in July by
more than half. The increased scattering shown in Fig. 11 from May to June is consistent with this observed average size
increase, but there is no decrease seen in July scattering. The large variability in both size and number for July makes
prediction of observed scattering increases very problematic. This suggests that when the ice is truly rotten and porous, and
the pores are very large, as was observed in July, that light scattering cannot be well represented by a simple evaluation of
average pore size and number density.
Early in the season, the larger scattering near the ice bottom likely reflects the higher brine content (larger and/or more
numerous brine inclusions) near the growth interface. The larger scattering near the top ice surface likely results from the
less organized ice structure that forms prior to the onset of congelation growth during initial ice formation. As the melt
season progresses, this uppermost portion of the ice has additional enhanced scattering due to the drainage of above-
freeboard ice and the eventual development of a surface scattering layer. The enhanced scattering at the top and bottom of
the ice results in a C-shaped profile, consistent with observed salinity profiles. This C-shape appears to dominate the profiles
for April, May and June, but the July sample appears to have no memory of the characteristic C-shape found earlier in the
season. Given the significant structural retexturing that occurred by July, this should not be surprising.
Laboratory optical measurements made analogously to the ones in this study were carried out for melting first-year sea ice in
the open pack (see *Light et al.*, 2015). That data set included little information about the temporal progression of the ice, as
no one location was sampled more than once. However, interior ice scattering coefficients between 0.1−0.3 cm$^{-1}$ were found
for that ice in June and July, and these values are comparable to what was found in this study.
In an effort to use light scattering measurements to inform our understanding of ice rotting processes, we monitored the
optical properties of natural ice samples as they melted. Since most of the May core had *in situ* temperature > -5 ℃, only
small changes in sample density and light scattering properties were observed until the ice warmed to -2 ℃ (Fig. 11, dashed
curve). The lab-rot core shows significantly enhanced scattering, although not as large as the naturally rotted ice. This was
viewed as a preliminary attempt to create rotten ice in the laboratory. Differences between ice rotted in air and floating in the
ocean would likely be the rate of rot, and the relative abundance of gas-filled pore space relative to liquid pore space.
Refractive index contrasts mean that gas pores scatter more effectively than brine filled pores; thus, lab-rotted samples were
flooded in order to best mimic in situ rotted ice.

**5. Conclusions**

As Arctic sea ice melts during the summer season, its microstructure, porosity, bulk density, salinity, and permeability
undergo significant evolution. *In situ* measurements of sea ice documented off the northern coast of Alaska in May, June,
and July, indicate that sea ice transitioned from having 4−10 ppt salinity in May to near zero salt content in July. The ice
became extremely porous, with porosity values exceeding 10 % through most of the depth of the ice compared to <10 % for
ice collected in May and June. Some July porosity values approached 50 % at places in the ice interior. Brine pockets in
rotten ice are few; the ice is essentially fresh in composition and characterized by large, visible voids and channels on the
order of several millimeters in diameter. These changes result from increased air temperature, ocean heat, and prolonged
exposure to sunlight and leave the ice with dramatically increased porosity, pore space with increased connectivity, and
increased capacity to backscatter light. These changes have potential implications for the structural integrity, permeability to
surface melt water as well as ocean water, light partitioning, habitability, and melting behavior of late summer ice.
Specifically, increased connectivity with the ocean may affect how material (e.g., dissolved and particulate material,
including biological organisms and their byproducts) is exchanged at the ice/ocean boundary. Subsequent surface meltwater
flushing may in turn effectively rinse these constituents from the ice, making this enhanced connectivity central to the
control of ice-associated constituents well into the summer season. Rotten ice is a very different physical and chemical
habitat for microbial communities than earlier-season ice.
Reductions in bulk density were observed to occur from values approximately $0.90-0.94$ g cm$^{-3}$ to values as low as 0.6 g cm$^{-3}$
$^{3}$. Pore spaces within this low density ice, however, were typically well connected to the ocean. This left the low-density
summer ice to generally have very small freeboard and frequently be flooded by ambient seawater. Finally, and significantly,
field observations stress the lack of structural integrity of this porous, fragile ice, indicating that thickness-based models of
ice behavior may not accurately predict the behavior of late-season sea ice.
In addition to sampling naturally rotted sea ice, we have also attempted to simulate the rotting process in the laboratory. Our
laboratory optics measurements suggest that natural samples extracted early in the season can be at least partially rotted in
the laboratory. To achieve ice that is as rotted and structurally compromised as was observed to occur in nature, the
absorption of solar radiation may be a necessary parameter. Sunlight is key to the formation of surface scattering layers at
the air−ice interface. In the lab, ice was permitted to rot in air, so any melt that was produced would quickly drain away. In
nature, the ice necessarily floats in its own melt, and this may be a critical difference in the way that heat is delivered to the
ice. Increases in melt season length may bring increased occurrence of rotten ice, and the timing and character of the
seasonal demise of sea ice may be related to the evolution of the ice microstructure.

**Data availability**

Data archived at NSF Arctic Data Center https://arcticdata.io/catalog/#view/doi:10.18739/A28C9R366

**Author Contribution**

Research concept and general research plan contributed by KJ, BL, MO. BL, KJ, MO, CF, and SC designed the study and
planned the fieldwork. BL, MO, KJ, SC, and CF conducted the fieldwork in 2015; BL, KJ, and SF conducted the fieldwork
in 2017. CF compiled and analyzed all field data. BL collected and analyzed all optical data. CF performed the microscopy
and SF analyzed the microstructure images. CF performed micro-CT measurements, and the micro-CT analyses were
designed and conducted by CF, SF, RL, and ZC. CF and BL prepared the manuscript with contributions from all co-authors.

**Competing Interests**

The authors declare no conflicts of interest.

**Acknowledgements**

This work was supported by NSF Award PLR-1304228 to KJ (lead PI), BL, MO. SF had additional support from a Mary
Gates Research Scholarship (UW). We thank Julianne Yip for help with sample collection and processing, Hannah DeLapp
for data organization, and Michael Hernandez for GIS help. The field campaign was successful as a result of the enterprising
support of the Ukpeaġvik Iñupiat Corporation Science staff and affiliates in Utqiaġvik. Logistical support was provided by
CH2M Hill Polar Services. The authors also appreciate the constructive reviews of Sønke Maus and an anonymous reviewer,
which served to improve this manuscript.

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

**Table Captions**
Table 1. Collected ice cores, sample locations, and ambient conditions. Measured local conditions are ranges of hourly
averages of meteoric data measured between 10:00–18:00 local time from the NOAA Earth System Research Laboratory
Barrow Atmospheric Baseline Observatory (BRW) 8 km NE of Utqiaġvik (71.3230° N, 156.6114°W;
https://www.esrl.noaa.gov/gmd/obop/brw/).

# Table 1

| Date & Transport | Measured local conditions | | Location | Ice type | Lat | Lon | Measured in situ conditions | | Core | State | Ice thickness (cm) | Core length (cm) | Freeboard Depth (cm) | % Below FB |
|---|---|---|---|---|---|---|---|---|---|---|---|---|---|---|
| 6 May 2015 | Temperature range | -12.3 – -10.7 °C | M-CS | Landfast | 71.37558 | -156.52810 | Air at 11:00 AM | -9 °C | M-CS-01 | | 147 | 147 | 12.0 | 92% |
| Snowmachine | Relative humidity | 87 – 90% | | | | | Snow 9 cm below surface | -7 °C | M-CS-02 | | 145 | | 9.5 | 93% |
| | Wind speed | 3.9 – 5.5 m/s | | | | | Sackhole fill | -10 °C | M-CS-03 | | 146 | 149 | 10.5 | 93% |
| | | | | | | | | | M-CS-04 | | 145 | 149 | 10.0 | 93% |
| | | | | | | | | | M-CS-05 | | 144 | 147 | 10.0 | 93% |
| | | | | | | | | | M-CS-06 | | 146 | 139 | 10.0 | 93% |
| | | | | | | | | | M-CS-07 | | 144 | 147 | 10.0 | 93% |
| | | | | | | | | | M-CS-08 | | 145 | 147 | 10.0 | 93% |
| | | | | | | | | | M-CS-09 | | 142 | 149 | 9.0 | 94% |
| | | | | | | | | | M-CS-10 | | 144 | 145 | 10.0 | 93% |
| | | | | | | | | | M-CS-11 | | 142 | 145 | 9.0 | 94% |
| | | | | | | | | | M-CS-12 | | 143 | 144 | 9.0 | 94% |
| | | | | | | | | | M-CS-13 | | 141 | 144 | 10.0 | 93% |
| | | | | | | | | | M-CS-14 | | 143 | 145 | 10.0 | 93% |
| | | | | | | | | | M-CS-15 | | 144 | 149 | 10.0 | 93% |
| | | | | | | | | | M-CS-16 | | 146 | 149 | 10.0 | 93% |
| | | | | | | | | | M-CS-17 | | 146 | | 10.0 | 93% |
| | | | | | | | | | M-CS-18 | | 146 | | 11.0 | 92% |
| | | | | | | | | | M-CS-19 | | 146 | | 11.0 | 92% |
| | | | | | | | | | M-CS-20 | | 149 | | 11.0 | 93% |
| | | | | | | | | | M-CS-21 | | 150 | | 12.0 | 92% |
| | | | | | | | | | M-CS-22 | | 149 | | 10.0 | 93% |
| 3 Jun 2015 | Temperature range | -3.0 – -1.8 °C | JN-CS | Landfast | 71.37581 | -156.52793 | Air at 12:00 PM | -1.6 °C | JN-CS-a | | 157 | 158 | 25.0 | 84% |
| ATV | Relative humidity | 88 – 94% | | | | | Seawater | 0.0 °C | JN-CS-b | | 149 | | 13.0 | 91% |
| | Wind speed | 4.9 – 6.1 m/s | | | | | | | JN-CS-01 | | 155 | 158 | 23.0 | 85% |
| | | | | | | | | | JN-CS-02 | | 155 | 158 | 22.0 | 86% |
| | | | | | | | | | JN-CS-03 | | 155 | 155 | 21.0 | 86% |
| | | | | | | | | | JN-CS-04 | | 154 | 156 | 22.0 | 86% |
| | | | | | | | | | JN-CS-05 | | 153 | 156 | 20.0 | 87% |
| | | | | | | | | | JN-CS-06 | | 153 | 152 | 20.0 | 87% |
| | | | | | | | | | JN-CS-07 | | 153 | 156 | 21.0 | 86% |
| | | | | | | | | | JN-CS-08 | | 152 | 152 | 22.0 | 86% |
| | | | | | | | | | JN-CS-09 | | 152 | 160 | 22.0 | 86% |
| | | | | | | | | | JN-CS-10 | | 152 | 155 | 22.0 | 86% |
| | | | | | | | | | JN-CS-11 | | 152 | 152 | 22.0 | 86% |
| | | | | | | | | | JN-CS-12 | | 152 | | 20.0 | 87% |
| | | | | | | | | | JN-CS-13 | | 151 | 151 | 21.0 | 86% |
| | | | | | | | | | JN-CS-14 | | | 155 | | |
| | | | | | | | | | JN-CS-15 | | | | | |
| | | | | | | | | | JN-CS-16 | | 155 | 159 | 21.0 | 86% |
| | | | | | | | | | JN-CS-17 | | 156 | 160 | 22.0 | 86% |
| | | | | | | | | | JN-CS-18 | | 157 | 160 | 22.0 | 86% |
| | | | | | | | | | JN-CS-19 | | 157 | 159 | 22.0 | 86% |
| | | | | | | | | | JN-CS-20 | | 156 | 160 | 21.0 | 86% |
| | | | | | | | | | JN-CS-21 | | 159 | 158 | 24.5 | 85% |
| | | | | | | | | | JN-CS-22 | | 155 | | 20.5 | 87% |
| | | | | | | | | | JN-CS-23 | | | | | |
| | | | | | | | | | JN-CS-24 | | | | | |

| Date & Transport | Measured local conditions | | Location | Ice type | Lat | Lon | Measured in situ conditions | | | Core | State | Ice thickness (cm) | Core length (cm) | Freeboard Depth (cm) | % Below FB |
|---|---|---|---|---|---|---|---|---|---|---|---|---|---|---|---|
| 3 Jul 2015 | Temperature range | 1.6 – 2.9 °C | JY3-L1 | ~15 m² white floe | 71.37836 | -157.11427 | Sackhole fill (10 cm deep) | -0.3 °C | 6 ppt | JY3-L1-01 | | 170 | 181 | 15.0 | 91% |
| Vessel 'Kimmialuk' | Relative humidity | 101% | | | | | Sackhole fill (35 cm deep) | -0.9 °C | 6 ppt | | | | | | |
| | Wind speed | 3.2 – 6.0 m/s | | | | | Seawater | 1.5 °C | 23 ppt | | | | | | |
| | | | JY3-L2 | ~2m² light brown floe | 71.38550 | -157.07941 | | | | JY3-L2-01 | | 150 | 151 | 16.0 | 89% |
| | | | | (drift ~2.5 km/hr NE) | | | | | | JY3-L2-02 | | 125 | 126 | 14.0 | 89% |
| | | | | | | | | | | JY3-L2-03 | | 107 | 91 | 6.0 | 94% |
| | | | | | | | | | | JY3-L2-04 | | 86 | 92 | 12.0 | 86% |
| | | | | | | | | | | JY3-L2-05 | | 109 | 112 | 8.0 | 93% |
| | | | | | | | | | | JY3-L2-06 | | 103 | 115 | 9.0 | 91% |
| | | | | | | | | | | JY3-L2-07 | | 116 | 116 | 14.0 | 88% |
| | | | | | | | | | | JY3-L2-08 | | 103 | 103 | 8.0 | 92% |
| | | | JY3-L3 | Large, heavily-ponded floe | 71.30055 | -155.63469 | | | | JY3-L3-01 | | 145 | 152 | 18.0 | 88% |
| 10 Jul 2015 | Temperature range | 1.4 – 2.5 °C | JY10 | Sediment-rich, heavily-ponded floe, 190 cm thick in non-ponded areas, broke up under light wave action | 71.44647 | -156.53116 | | | | JY10-01 | ponded | 81 | 107 | -17.5 | |
| Vessel 'Jenny Lee' | Relative humidity | 0.99 | | | | | | | | JY10-02 | ponded | 86 | 99 | | |
| | Wind speed | 3.9 – 4.7 m/s | | | (drift ~ 1 km/hr NNW) | | | | | JY10-03 | ponded | 82 | 87 | -17.5 | |
| | | | | | | | | | | JY10-04 | ponded | 90 | 82 | | |
| | | | | | | | | | | JY10-05 | ponded | 85 | 90 | | |
| | | | | | | | | | | JY10-06 | ponded | 90 | 92 | | |
| | | | | | | | | | | JY10-07 | ponded | 85 | 88 | | |
| | | | | | | | | | | JY10-08 | ponded | 90 | 85 | | |
| | | | | | | | | | | JY10-09 | ponded | | 85 | | |
| | | | | | | | | | | JY10-10 | ponded | 58 | 68 | -18.0 | |
| | | | | | | | | | | JY10-11 | ponded | 83 | 92 | | |
| | | | | | | | | | | JY10-12 | ponded | 80 | 95 | | |
| | | | | | | | | | | JY10-13 | ponded | 87 | | | |
| | | | | | | | | | | JY10-14 | ponded | 79 | 81 | | |
| | | | | | | | | | | JY10-15 | ponded | 80 | 91 | | |
| 11 Jul 2015 | Temperature range | 0.7 – 2.5 °C | JY11 | Sediment-poor, white, ponded floe | 71.40163 | -156.02647 | Air | 1.0 – 1.3 °C | | JY11-01 | ponded | 89 | 93 | -8.8 | |
| Vessel 'Jenny Lee' | Relative humidity | 96 – 101% | | | | | (drift ~ 2 km/hr NW) | Melt pond | -0.3 °C | JY11-02 | ponded | 86 | 85 | -10.0 | |
| | Wind speed | 4.4 – 5.3 m/s | | | | | | | | JY11-03 | ponded | 80 | 89 | -10.0 | |
| | | | | | | | | | | JY11-04 | ponded | 70 | 65 | -12.5 | |
| | | | | | | | | | | JY11-05 | ponded | 90 | 95 | -12.5 | |
| | | | | | | | | | | JY11-06 | ponded | 90 | 80 | -12.5 | |
| | | | | | | | | | | JY11-07 | ponded | 90 | 95 | | |
| | | | | | | | | | | JY11-08 | ponded | 84 | 89 | | |
| | | | | | | | | | | JY11-09 | ponded | 70 | 74 | | |
| | | | | | | | | | | JY11-10 | ponded | 65 | 72 | -15.0 | |
| | | | | | | | | | | JY11-11 | ponded | 83 | 80 | | |
| | | | | | | | | | | JY11-12 | ponded | 83 | 88 | -7.5 | |
| | | | | | | | | | | JY11-13 | ponded | 83 | | | |
| | | | | | | | | | | JY11-14 | ponded | 83 | | | |
| | | | | | | | | | | JY11-15 | ponded | 83 | | | |
| | | | | | | | | | | JY11-16 | ponded | 83 | | | |
| | | | | | | | | | | JY11-17 | ponded | 80 | | | |
| | | | | | | | | | | JY11-18 | ponded | | | | |
| | | | | | | | | | | JY11-19 | ponded | 85 | 74 | | |
| | | | | | | | | | | JY11-20 | ponded | 71 | 70 | | |
| | | | | | | | | | | JY11-21 | ponded | | 86 | | |
| | | | | | | | | | | JY11-22 | ponded | 64 | 69 | | |

| Date & Transport | Measured local conditions | | Location | Ice type | Lat | Lon | Measured in situ conditions | | | Core | State | Ice thickness (cm) | Core length (cm) | Freeboard Depth (cm) | % Below FB |
|---|---|---|---|---|---|---|---|---|---|---|---|---|---|---|---|
| 14 Jul 2015 | Temperature range | 2.7 – 3.6 °C | JY14-L3 | Large floe with thin ice | 71.39465 | -156.25842 | | | | JY14-L3-01 | ponded | 54 | 53 | -7.5 | |
| Vessel 'Jenny Lee' | Relative humidity | 97 – 101% | | | | | | | | JY14-L3-02 | ponded | 68 | 80 | -7.5 | |
| | Wind speed | 4.2 – 5.5 m/s | | | | | | | | JY14-L3-03 | ponded | 72 | 84 | -10.0 | |
| | | | | | | | | | | JY14-L3-04 | ponded | 78 | 91 | -12.5 | |
| | | | | | | | | | | JY14-L3-05 | | 100 | 111 | | |
| | | | | | | | | | | JY14-L3-06 | | 115 | 121 | | |
| | | | | | | | | | | JY14-L3-07 | | 139 | 146 | | |
| | | | | | | | | | | JY14-L3-08 | | | 119 | | |
| | | | | | | | | | | JY14-L3-09 | | | 164 | | |
| | | | | | | | | | | JY14-L3-10 | | | 118 | | |
| | | | | | | | | | | JY14-L3-11 | | | 115 | | |
| | | | | | | | | | | JY14-L3-12 | ponded | 53 | 48 | | |
| | | | | | | | | | | JY14-L3-13 | ponded | | 47 | | |
| | | | | | | | | | | JY14-L3-14 | ponded | | 47 | -10.0 | |
| | | | | | | | | | | JY14-L3-15 | ponded | | 64 | | |
| | | | | | | | | | | JY14-L3-16 | ponded | | 27 | -0.5 | |
| | | | JY14-L4 | Uniformly thin, flat ice | 71.41843 | -156.23720 | | | | JY14-L4-01 | | | 80 | | |
| | | | | | | | | | | JY14-L4-02 | | | 83 | | |
| | | | | | | | | | | JY14-L4-03 | ponded | | 72 | -5.0 | |
| | | | | | | | | | | JY14-L4-04 | ponded | | 69 | -5.0 | |
| | | | | | | | | | | JY14-L4-05 | | | 81 | | |
| | | | | | | | | | | JY14-L4-06 | ponded | | 70 | -5.0 | |
| 13 Jul 2017 | Temperature range | 4.4 –8.9 °C | JY13-L1 | Uniformly thin, flat ice | | | Air | ~ 8 –11 °C | | JY13-L1-01 | | 40 | | | |
| Vessel 'Crescent Island' | Relative humidity | 84 – 96% | JY13-L2 | Uniformly thin, flat ice | 71.50761 | -156.17486 | | | | JY13-L2-01 | | 60 – 70 | | | |
| | Wind speed | 4.0 – 8.3 m/s | | | | | | | | JY13-L2-02 | | (approx. range for all cores) | | | |
| | | | | | | | | | | JY13-L2-03 | | | | | |
| | | | | | | | | | | JY13-L2-04 | | | | | |
| | | | JY13-L3 | Uniformly thin, flat ice | | | | | | JY13-L3-01 | | | | | |
| | | | | | | | | | | JY13-L4-01 | | | | | |
| | | | JY13-L4 | Uniformly thin, flat ice | | | | | | | | | | | |
| | | | JY13-L5 | Uniformly thin, flat ice | 71.49875 | -156.18888 | Seawater surface | 4.1 °C | 23.0 ppt | JY13-L5-01 | | | | | |
| | | | | | | | Seawater 1.5 m depth | 4.3 °C | 25.0 ppt | JY13-L5-02 | | | | | |
| | | | | | | | Seawater 3.0 m depth | 4.8 °C | 29.5 ppt | | | | | | |
| 17 Jul 2017 | Temperature range | 5.9 – 8.3 °C | JY17-L1 | | 71.52508 | -156.02848 | | | | JY17-L1-01 | | 62 – 110 | | | |
| Vessel 'Doctor Island' | Relative humidity | 101 – 103% | | | | | | | | JY17-L1-02 | | (range for all cores) | | | |
| | Wind speed | 4.8 – 6.5 m/s | | | | | | | | JY17-L1-03 | | | | | |
| | | | | | | | | | | JY17-L1-04 | | | | | |
| | | | | | | | | | | JY17-L1-05 | | | | | |
| | | | | | | | | | | JY17-L1-06 | | | | | |
| | | | | | | | | | | JY17-L1-07 | | | | | |
| | | | | | | | | | | JY17-L1-08 | | | | | |