# Peer review of "Physical and optical characteristics of heavily melted "rotten" Arctic sea ice"

_The Cryosphere, 2018_

## Referee Comment (RC1) · Anonymous Referee #1 · 5 Sep 2018

General comments

This manuscript provided a good and detailed investigation on the "rotten sea ice" during the melt season of Arctic. Physical and optical properties of such ice were measured through in-situ investigations and discussed here. The potential readers can get a lot of useful information from this manuscript, but a problem also exist accordingly: a very clear subject is absent throughout the manuscript. I understand that the data presented in the manuscript are good, but I think the structure of manuscript should be improved to focus on one or two scientific topics, for example, the difference between rotten ice and often-studied summer ice. And then the whole content will look like a good paper rather than a data report.

The manuscript is well-written, and the quality of the presentation is good. However,

the manuscript can be further improved upon addressing the points above and below.

Specific comments

1. Rotten ice is not an often-used word in previous publications, so can you provide a very clear definition first in the introduction section?

2. In section 2.1, there are titles for the subsection: "2.1.1 May, 2.1.2 June, 2.1.3 July 2015, 2.1.4, July 2017". Then what is the year for the first two titles?

3. Figures 1, 2, and 3, gave some in-situ pictures, but they seem to be somewhat repeated. So please consider to remove some of them and leave the important ones.

4. In section 2.2.3, line 178, manual thresholding gave more reliable results than automatic thresholding. I understand this because automatic thresholding cannot handle all situations we met, but manual thresholding is very sensitive to the person who perform the image segmentation. So how to evaluate the error of manual thresholding?

5. In section 2.3, a method to measure inherent optical properties of sea ice in laboratory is introduced, but the process is still a little difficult to understand by readers who are not so familiar with optics. Can you add a figure here to explain the laboratory method?

6. In section 4.1, yes, the equation of Cox and Weeks [1983] is valid only for ice temperature less than -2°C, but Lepparanta and Manninen [1988] has setup a new equation to solve the problem as temperature more than -2°C. The authors should cite the paper Leppäranta M, Manninen T. 1988. The brine and gas content of sea ice with attention to low salinities and high temperatures. Finnish Institute for Marine Research, Internal Report, 1988(2): 14.

7. In section 4.2, line 478. Increasing in ice scattering seemed to be a result of changes of ice microstructure, so can we give some quantitative results here because both optical and physical parameters were measured in this study?

Technical corrections

1. Line 44, Eicken et al. [2002] noted. . .

---

## Referee Comment (RC2) · S. Maus (Referee) · 14 Sep 2018

**Review of manuscript tc-2018-141**
by Sönke Maus

This is a review of the manuscript *Physical and optical characteristics of heavily melted 'rotten' Arctic sea ice*  by C. M. Frantz et al.

Below I cite from the Cryosphere Discussions manuscript tc-2018-141 in *italic font*.

**I   Summary**

The paper presents an analysis of the physical and optical properties of heavily melted Arctic first year ice. At present very little is known about the physical properties of such ice that plays an important role for, among other processes, radiative transfer. The topic is thus absolutely worth publishing in The Cryosphere. Beyond standard measurements of physical properties on bulk ice samples (temperature,salinity,density) also an analysis of 3-d tomographic observations of the microstructure is presented. The article is well written and structured into the sections 1.Introduction, 2. Materials and methods, 3. Results, 4. Discussion and 5.Conclusions.

I find the manuscript interesting and well written. New observations of sea ice properties from the onset of melt to its rotten state are well presented and analysed in terms of radiative transfer. I found two weaknesses, that should be straightforward to address, which would improve the quality of the manuscript.

1. As described in more detailed comments below, there are some issues with the sample treatment, especially the flooding of samples with brine and DMP, that should be addressed. When a centrifuged sample is re-filled with a liquid, it is rather probable that part of the pore space is not filled, creating artificial air pockets or bubbles. The creation of extra bubbles may affect two of the microstructure metrics addressed by XRT, and it may also influence the interpretation in terms of scattering model results. While the authors mention the aspect of bubble formation during flooding, they could have provided a quantitative evaluation. The question could for example be addressed by a more detailed analysis of the micro-CT derived different open and closed porosity fractions (air, brine, injected DMP), rather than discussing just total porosity.

2. Anisotropy of pores and inclusions is a rather fundamental aspect of sea ice microstructure, and it is likely to play a role for many processes as well as radiative transfer (Katlein et al., 2014). Anisotropy of sea ice microstructure is not well documented yet and it is an important contribution of the manuscript to address it. However, the presentation of anisotropy in the manuscript is inconsistent, see below. To a certain degree this inconsistency appears to come from adopting the definition and determination of anisotropy as proposed by Lieb-Lappen et al. (2017).

I would like to recommend the manuscript for publication, after these too aspects have been addressed.

**II Specific comments**

**1. Introduction**

P 2, L36-42 –> *In general, the connectivity of an ice cover is known to...* –> I would put the paragraph on ice dynamics (L50-59) here together with the mentioned processes, and rather join the sentence on permeability (L40-42) with the next paragraph (L43-49) on this topic.

P 2, L36 –> *Increases in ice permeability result in an increase in the amount of surface meltwater...* –> if the amount increases may depends on other factors, so better use 'flow rate'

P 2, L36 –> *As a result of the notable connectivity of its microstructure* –> Better 'connectivity of its pore space'

**2. Materials and Methods**

P 3-8 –> This section describes the samples taken on three sampling dates. It would be helpful for the reader to summarise the characteristics (date,thickness, air temperature, ice salinity, freeboard) in a table.

P 8, L 148 –> I assume that pack volume was estimated for density measurements. Could you estimate the accuracy of these measurments?

P 8, L 150 –> The mentioned accuracy seems too good for a hand-held instrument. According to my information (handbook) the YSI Model 30 has a salinity accuracy $\pm$ 2 %, not $\pm$ 0.2 %.

P 8, L 153 –> To which thickness were thin sections microtomed? Could you mention a reference?

P 9, L 161-163 –> The working temperatures were -5, -2 and -1℃, and the same storage temperatures were chosen. However centrifuging was performed at the same temperature of -5℃. This may effect the microstructure considerably (e.g. for -1 ℃ brine volume might decrease by a factor of 4). Can you comment on this effect? As you mention, that the brine has been collected for further analysis, you can do so by asking: does the brine salinity correspond to the equilibrium brine salinity at the working temperature?

P 9, L 165-170 –> What is the reason to use DMP casting on the centrifuged images? This clearly complicates the analysis of XRT images, but an advantage is not mentioned. Note also that, as for the flooding with brine, flooding with DMP is likely to entrap air and thus overestimate the air porosity.

P 11, L229-232 –> I assume that the described flooding requires samples to be placed into a box or tube, which raises some questions: Were samples taken out of the flooding tube again for optical measurements? Also, I have myself attempted such flooding of

centrifuged samples, but never managed to refill the original pore space - there are always pores that are not refilled. Do you have data to assess this question as for the DMP? E.g. a XRT-scan?

P 11, L239 –> The drainage in the laboratory would produce 'rotten' ice with a lot of air voids, while in the field ice may 'rot' differently, with internal melting increasing the brine/liquid content. As air voids are expected to be better scatterers, this difference should be mentioned and addressed in the discussion of Figure 10, see below.

**3. Results**

P 14, L289 –> How were the relative measurement errors for density calculated?

P 16, L322-325 –> The median is often a better description of a characteristic pore scale than the mean. It would be very helpful if you could plot your size distributions/histograms below the images in Fig.7 .

P 19, L364-366 –>My experience shows that the ratio of centrifuged to entrapped brine is typically in the range 0.5-4, with a value of 2 being most representative around a porosity of 0.1. So far data are limited, yet results are similar for young and old ice, showing that the ratio decreases with decreasing porosity (Maus et al., 2011, 2015). I therefore recommend to separately plot the relationship between open/closed ratio and total porosity. Doing so, I would prefer to plot the information as a fraction of open porosity to total porosity, rather than open porosity to closed porosity. The latter may diverge and makes it difficult to find a good plot scaling. There are also other arguments to do so, if one wants to interpret the results in terms of percolation theory.

The open to closed porosity ratio in this study may be biased by two factors: on the one hand, the DMP flooding may create artifical air bubbles. On the other hand are certain fractions of air bubbles and in particular disconnected brine inclusions not detected with the effective resolution of the micro-CT. The large values of open/closed porosity ratios (10-100) may therefore be in error. How much large could this error be? Could you address the question, how much artificial closed air pores the DMP intrusion may generate? This could be done by distinguishing between open and closed pores for air on the one hand and and brine+DMP on the other hand.

P 20, L378-388 –> The anisotropy measure from Lieb-Lappen et al. (2017) is used here. These authors define it this way (page 28, upper right paragraph):  *A polar plot encompassing all the mean intercept lengths is created by creating an ellipsoid with boundaries defined by the mean intercept length for each direction. Any given ellipsoid can be characterized by a matrix, and the eigenvalues for this matrix are calculated, which correspond to the lengths of the semi-major and semi-minor axes. The ratio of the largest to smallest eigenvalues then provides a metric for the degree of anisotropy, with 0 representing a perfectly isotropic object and 1 representing a completely anisotropic object..* The authors do not give any formula beyond this description, neither do they refer to any publication about (the apparently applied) mean intercept method in microstrcture analysis. There seems to be an error here, because when anisotropy is projected to the range 0-1, the ratio of minor to major axis length should the the correct definition. Also, based on the

definition of anisotropy as an axis length ratio, it would be vice versa to the description in this paper and in Lieb-Lappen et al. (2017): a value of 1 would present a perfectly isotropic object and a value of 0 an infinitely long anisotropic pore.

I think therefore that the whole description of anisotropy should be checked. It is actually intuitively surprising to find the highest anisotropy in the mid horizon (as the authors as well as Lieb-Lappen et al. (2017) describe), rather than near the bottom of sea ice, where brine channels and seawater are well connected.

It is finally worth mentioning that anisotropy, if defined as minor to major axis ratio in this way, would be a problematic measure when considering through-sample brine channels. For this case the major axis is limited by the sample length and the measure would be size-dependent.

P 22, L424 –> *Submerged cores appear to have more porous ice structure.* –> Could this be supported by some of the XRT masurements? Proposing this and the following from only the photographs sounds a bit speculative.

P 22, L424 –> *By June, the salinity profile shows freshening at the ice 434 bottom, likely associated with the onset of bottom ablation.* –> Another explanation could be, as the authors proposeed earlier, that this warmer ice has wider pores and looses much more brine during sampling. Fig. 9c actually supports this. If true, then the ice may only have an apparently lower salinity. This question could be addressed by a closer look into the XRT images.

**4. Discussion**

P 22, L424 –> *In particular, the micro-CT work is useful for sampling much larger sample volumes, and thus central for estimating size and number distributions for the July 465 ice.* –> This claim raises several questions: 1. How may the number density of inclusions be effected by the DMP flooding process? 2. The micro-CT measurements were limited to a voxel size of 280 micron - how can optical and micro-CT number estimates be combined and compared?

P 23, L448-451 –> *Normally, sea ice with significantly smaller bulk density would be expected to float higher in the water and thus have larger freeboard. But the density reductions that occur during advanced melt result from large void spaces within the ice that are typically in connection with the ocean. As a result, such ice can have small freeboard, even if total ice thickness is still relatively large.* –> I would interpret the low densities rather due to rapid brine drainage during sampling, creating apparent low densities. This question should be further adressed. Again, the micro-CT observations may be used here for clarification, by splitting them up into brine, air and DMP porosities.

P 24, L488-491 –> *Our findings are consistent with those of Jones et al. (2012), which used cross-borehole DC resistivity tomography to observe increasing anisotropy of brine structure during spring warming. In that work, the brine phase was found to be connected both vertically and horizontally and the dimensions of vertically oriented brine channels gradually increased as the ice warmed.* –> I agree, this is consistent, and it is what one intuitively would expect. However, in the results section (P 20, L378-388) you say something different. This again underlines the above mentioned inconsistency in the anisotropy description from Lieb-Lappen et al. (2017).

P 24, L492-493 –> As you have results from microscopy and micro-CT you could quantify this results. E.g. plot both size distributions in a histogram. This would indicate to what degree the methods are comparable in the overlapping regime, and what resolution a CT-Scanner should have.

P 25, L518-522 –> As mentioned above, the drainage in the lab would produce 'rotten' ice with a lot of air voids, while in the field ice may 'rot' differently, increasing mostyl the brine porosity. Could you comment on the question, to what degree the applied model treats air and brine scattering differently?

**5. Discussion**

P 26, L538-542 –> See above note: I would interpret the low densities rather due to rapid brine drainage during sampling, creating apparent low densities.

P 26, L548 –> *critical difference* –> In terms of....scattering?

**III   Figures and References**

Fig. 6 –> It would be nice to have the measured freeboard indicated in the different profiles.
   Also an easy-to-see distinguishment of ponded and unponded ie would be helpful.

P 25, L520 –> *Fig. 10, dashed curve* –> the 'dashed' is difficult to see

**References**

Katlein, C., Nicolaus, M., Petrich, C., 2014. The anisotropic scattering coefficient of sea ice. J. Geophys. Res. Oceans 119 (119), 842–855.

Lieb-Lappen, R., Golden, E., Obbard, R., 2017. Metrics for interpreting the microstructure of sea ice using x-ray micro-computed tomography. Cold Reg. Sci. Technol. 138, 24–35.

Maus, S., Becker, J., Leisinger, S., Matzl, M., Schneebeli, M., Wiegmann, A., June 2015. Oil saturation of the sea ice pore space. In: Proceedings - Port and Ocean Engineering under Arctic Conditions, Trondheim, Norway. POAC, 12 pp.

Maus, S., Haase, S., Büttner, J., Huthwelker, T., Schwikowski, M., Vähätalo, A., Enzmann, F., 2011. Ion fractionation in young sea ice from Kongsfjorden, Svalbard. Annals Glaciol. 52 (57), 301–310.

---

## Author Comment (AC1) · 29 Nov 2018

**Physical and optical characteristics of heavily melted "rotten" Arctic sea ice**
**Author response to reviewer comments**

We would like to sincerely thank the reviewers for their feedback, which we feel substantially improved the manuscript. We respond to their points here, and outline the changes we have made to the manuscript.

**Reviewer #1**

*This manuscript provided a good and detailed investigation on the "rotten sea ice" during the melt season of Arctic. Physical and optical properties of such ice were measured through in-situ investigations and discussed here. The potential readers can get a lot of useful information from this manuscript, but a problem also exist accordingly: a very clear subject is absent throughout the manuscript. I understand that the data presented in the manuscript are good, but I think the structure of manuscript should be improved to focus on one or two scientific topics, for example, the difference between rotten ice and often-studied summer ice. And then the whole content will look like a good paper rather than a data report. The manuscript is well-written, and the quality of the presentation is good. However, the manuscript can be further improved upon addressing the points above and below.*

We thank the reviewer for this helpful review, and appreciate feedback about the style of this manuscript. We have edited the manuscript, in particular, aiming to help the reader distinguish "rotten ice" as a distinct form of summer ice.

*Specific comments*

*1. Rotten ice is not an often-used word in previous publications, so can you provide a very clear definition first in the introduction section?*

By characterizing "rotten" ice, the paper serves to provide a quantitative definition. However, we recognize the circular nature of providing a definition of something undefined and have added a descriptive definition to the end of the introduction.

*2. In section 2.1, there are titles for the subsection: "2.1.1 May, 2.1.2 June, 2.1.3 July 2015, 2.1.4, July 2017". Then what is the year for the first two titles?*

We added years to the subtitles.

*3. Figures 1, 2, and 3, gave some in-situ pictures, but they seem to be somewhat repeated. So please consider to remove some of them and leave the important ones.*

Fig. 1 shows satellite views for the four sampling periods, Fig. 2 shows surface panoramas, Fig. 3 shows specific sample locations. We deleted panel (c) in Fig. 2, since we agree it was accessory. Because this report serves to define and officially document rotten ice, we feel it is useful to show the different visual examples in Figure 3.

*4. In section 2.2.3, line 178, manual thresholding gave more reliable results than automatic thresholding. I understand this because automatic thresholding cannot handle all situations we met, but manual thresholding is very sensitive to the person who perform the image segmentation. So how to evaluate the error of manual thresholding?*

The phase selection was clarified in the methods, and the following sentence was added: "A preliminary sensitivity analysis indicated that manual thresholding by a single analyst was found to give more reliable

results than automated thresholding methods due to relatively large variability in brightness and contrast in reconstructed images as well as poor brightness separation between the ice and DMP phases."

*5. In section 2.3, a method to measure inherent optical properties of sea ice in laboratory is introduced, but the process is still a little difficult to understand by readers who are not so familiar with optics. Can you add a figure here to explain the laboratory method?*

We agree with this suggestion, and have added a new figure (now Fig. 4) to help make this laboratory technique easier to visualize.

*6. In section 4.1, yes, the equation of Cox and Weeks [1983] is valid only for ice temperature less than -2∘C, but Lepparanta and Manninen [1988] has setup a new equation to solve the problem as temperature more than -2∘C. The authors should cite the paper Leppäranta M, Manninen T. 1988. The brine and gas content of sea ice with attention to low salinities and high temperatures. Finnish Institute for Marine Research, Internal Report, 1988(2): 14.*

Added.

*7. In section 4.2, line 478. Increasing in ice scattering seemed to be a result of changes of ice microstructure, so can we give some quantitative results here because both optical and physical parameters were measured in this study?*

We have added the following text to the discussion: "Relative increases in the scattering would be expected to scale by the inclusion number density multiplied by the square of the effective inclusion radius (see *Light et al.*, 2003). Observed mean inclusion sizes increased from average May size of 80 μm to average June size of 221 μm to average July size of 3 mm. Observed number densities decreased from 32 mm$^{-3}$ (May) to 19 mm$^{-3}$ (June) to 0.01 mm$^{-3}$ (July). These changes correspond to relative scattering coefficient magnitude changes of 1: 4.5: 0.4, which would predict a scattering coefficient increase from May to June by a factor of 4.5, and a decrease in July by more than half. The increased scattering shown in Fig. 11 from May to June is consistent with this observed average size increase, but there is no decrease seen in July scattering. The large variability in both size and number for July makes prediction of observed scattering increases very problematic. This suggests that when the ice is truly rotten and porous, and the pores are very large, as was observed in July, that light scattering cannot be well represented by a simple evaluation of average pore size and number density."

*8. Technical corrections 1. Line 44, Eicken et al. [2002] noted. . .*

Corrected, thanks.

**Reviewer #2 S. Maus**

We thank the reviewer for this thorough, thoughtful, and constructive review.

**I Summary**
*The paper presents an analysis of the physical and optical properties of heavily melted Arctic first year ice. At present very little is known about the physical properties of such ice that plays an important role for, among other processes, radiative transfer. The topic is thus absolutely worth publishing in The Cryosphere. Beyond standard measurements of physical properties on bulk ice samples (temperature,salinity,density) also an analysis of 3-d tomographic*

*observations of the microstructure is presented. The article is well written and structured into the sections 1.Introduction, 2. Materials and methods, 3. Results, 4. Discussion and 5.Conclusions.*

*I find the manuscript interesting and well written. New observations of sea ice properties from the onset of melt to its rotten state are well presented and analysed in terms of radiative transfer. I found two weaknesses, that should be straightforward to address, which would improve the quality of the manuscript.*

*1. As described in more detailed comments below, there are some issues with the sample treatment, especially the flooding of samples with brine and DMP, that should be addressed. When a centrifuged sample is re-filled with a liquid, it is rather probable that part of the pore space is not filled, creating artificial air pockets or bubbles. The creation of extra bubbles may affect two of the microstructure metrics addressed by XRT, and it may also influence the interpretation in terms of scattering model results. While the authors mention the aspect of bubble formation during flooding, they could have provided a quantitative evaluation. The question could for example be addressed by a more detailed analysis of the micro-CT derived different open and closed porosity fractions (air, brine, injected DMP), rather than discussing just total porosity.*

We treated samples collected in the field with DMP in an effort to preserve structure and to be consistent with work elucidating the structure of frozen samples done previously with snow (Schneebli group, e.g., Heggli *et al* 2009). When XRT scans and CT analyses were done, we realized that flooding the sample with DMP had introduced artifacts. As a result, we decided to base analyses in the manuscript on the total porosity (air + brine + DMP) instead of separately. Due to the problematic nature of DMP flooding, we believe it is best to focus on this "not ice"/pore fraction instead of attempting to make sense of patterns of DMP distribution, air inclusions, etc. We added language to the methods section to clarify this.

In the future, we would not recommend DMP casting as a technique for sea ice, and we added to the discussion of the issues with DMP flooding in the revised manuscript.

*2. Anisotropy of pores and inclusions is a rather fundamental aspect of sea ice microstructure, and it is likely to play a role for many processes as well as radiative transfer (Katlein et al., 2014). Anisotropy of sea ice microstructure is not well documented yet and it is an important contribution of the manuscript to address it. However, the presentation of anisotropy in the manuscript is inconsistent, see below. To a certain degree this inconsistency appears to come from adopting the definition and determination of anisotropy as proposed by Lieb-Lappen et al. (2017).*

See response below.

*I would like to recommend the manuscript for publication, after these too aspects have been addressed.*

**II Specific comments**

**1. Introduction**

*P 2, L36-42 –> In general, the connectivity of an ice cover is known to... –> I would put the paragraph on ice dynamics (L50-59) here together with the mentioned processes, and rather join the sentence on permeability (L40-42) with the next paragraph (L43-49) on this topic.*
Good suggestion, thank you.

*P 2, L36 –> Increases in ice permeability result in an increase in the amount of surface meltwater... –> if the amount increases may depends on other factors, so better use 'flow rate'*
Yes, correct, change made.

*P 2, L36 –> As a result of the notable connectivity of its microstructure –> Better 'connectivity of its pore space'*
Yes, agreed.

**2. Materials and Methods**

*P 3-8 –> This section describes the samples taken on three sampling dates. It would be helpful for the reader to summarise the characteristics (date,thickness, air temperature, ice salinity, freeboard) in a table.*
A summary table has been added, thank you for the suggestion.

*P 8, L 148 –> I assume that puck volume was estimated for density measurements. Could you estimate the accuracy of these measurments?*
Accuracy of this method is limited by determination of volume. Multiple diameter and thickness measurements were averaged, and used to calculate puck volumes. Density errors were calculated by propagating errors (dominated by the variability in volume estimation, especially late in the season where pucks were uneven and sometimes crumbly). We have added an explicit statement describing the error propagation, and this is further discussed in the results section.

*P 8, L 150 –> The mentioned accuracy seems too good for a hand-held instrument. According to my information (handbook) the YSI Model 30 has a salinity accuracy _ 2 %, not _ 0.2 %.*
Yes, 2% is correct, thank you for catching that.

*P 8, L 153 –> To which thickness were thin sections microtomed? Could you mention a reference?*
2 mm is now specified for sample thickness in the text

*P 9, L 161-163 –> The working temperatures were -5, -2 and -1., and the same storage temperatures were chosen. However centrifuging was performed at the same temperature of -5.. This may effect the microstructure considerably (e.g. for -1 . brine volume might decrease by a factor of 4). Can you comment on this effect? As you mention, that the brine has been collected for further analysis, you can do so by asking: does the brine salinity correspond to the equilibrium brine salinity at the working temperature?*

Indeed, the microstructure of sea ice would be expected to be highly sensitive to such temperature changes. In this case, however, we expect the most severe changes occurred when the ice samples were cut and removed from the ice cover and transported back to the freezer. During warmer sampling days (in July), a considerable amount of liquid was drained from the samples during this process (as was evidenced by the accumulated liquid in the bags that had to be drained prior to putting samples in the freezer). Despite this, cores looked visually similar in terms of different regions of scatter, porosity, granularity, transparency, etc. to when they were collected in the field, qualitatively indicating some structural consistency.

Furthermore, samples were held at their working temperature until immediately before centrifuging. The centrifuging process was done rapidly and it was assumed that the samples had enough thermal inertia to approximately hold their temperature. We have added this information to the text.

*P 9, L 165-170 –> What is the reason to use DMP casting on the centrifuged images? This clearly complicates the analysis of XRT images, but an advantage is not mentioned. Note also that, as for the flooding with brine, flooding with DMP is likely to entrap air and thus overestimate the air porosity.*

See response to question #1 above.

*P 11, L229-232 –> I assume that the described flooding requires samples to be placed into a box or tube, which raises some questions: Were samples taken out of the flooding tube again for optical measurements? Also, I have myself attempted such flooding of centrifuged samples, but never managed to refill the original pore space - there are always pores that are not refilled. Do you have data to assess this question as for the DMP? E.g. a XRT-scan?*

Optical measurements were made on the ice sample while it was still flooded. The optical chamber is water tight, and that has now been made explicit in the description of the optical measurements. Additionally, it is entirely possible that the flooding process was not perfect. It was however noted that the visual appearance of the samples indicated dramatic reduction in backscatter when the samples were flooded, indicating that, even if not perfect, the flooding had a significant effect.

*P 11, L239 –> The drainage in the laboratory would produce 'rotten' ice with a lot of air voids, while in the field ice may 'rot' differently, with internal melting increasing the brine/liquid content. As air voids are expected to be better scatterers, this difference should be mentioned and addressed in the discussion of Figure 10, see below.*
Addressed at very end of section 4.

**3. Results**
*P 14, L289 –> How were the relative measurement errors for density calculated?*
See prior question about density error estimates.

*P 16, L322-325 –> The median is often a better description of a characteristic pore scale than the mean. It would be very helpful if you could plot your size distributions/ histograms below the images in Fig.7 .*

This was an excellent suggestion. Histograms have been added to Figure 8 (was Figure 7), which indicate a clear shift in pore size distribution.

*P 19, L364-366 –>My experience shows that the ratio of centrifuged to entrapped brine is typically in the range 0.5-4, with a value of 2 being most representative around a porosity of 0.1. So far data are limited, yet results are similar for young and old ice, showing that the ratio decreases with decreasing porosity (Maus et al., 2011, 2015). I therefore recommend to separately plot the relationship between open/closed ratio and total porosity. Doing so, I would prefer to plot the information as a fraction of open porosity to total porosity, rather than open porosity to closed porosity. The latter may diverge and makes it difficult to find a good plot scaling. There are also other arguments to do so, if one wants to interpret the results in terms of percolation theory.*
*The open to closed porosity ratio in this study may be biased by two factors: on the one hand, the DMP flooding may create artifical air bubbles. On the other hand are certain fractions of air bubbles and in particular disconnected brine inclusions not detected with the effective resolution of the micro-CT. The large values of open/closed porosity ratios (10-100) may therefore be in error. How much large could this error be? Could you address the question, how much artificial closed air pores the DMP intrusion may generate? This could be done by distinguishing between open and closed pores for air on the one hand and and brine+DMP on the other hand.*

We have replaced the open/closed ratio panel with an open/total porosity volume panel in Figure 10 (was Figure 9), which we agree better-illustrates the changes in the character of pores. We also redid some text in the interpretation to reflect this improved metric.

With regard to the potential for artefacts noted: values for open, closed, and total pores reported are from the ice-only fraction, so the pores include air, brine, and DMP phases. The introduction of air with DMP, therefore, is not critical.

*P 20, L378-388 –> The anisotropy measure from Lieb-Lappen et al. (2017) is used here. These authors define it this way (page 28, upper right paragraph): A polar plot encompassing all the mean intercept lengths is created by creating an ellipsoid with boundaries defined by the mean intercept length for each direction. Any given ellipsoid can be characterized by a matrix, and the eigenvalues for this matrix are calculated, which correspond to the lengths of the semi-major and semi-minor axes. The ratio of the largest to smallest eigenvalues then provides a metric for the degree of anisotropy, with 0 representing a perfectly isotropic object and 1 representing a completely anisotropic object.. The authors do not give any formula beyond this description, neither do they refer to any publication about (the apparently applied) mean intercept method in microstrcture analysis. There seems to be an error here, because when anisotropy is projected to the range 0-1, the ratio of minor to major axis length should the the correct definition. Also, based on the definition of anisotropy as an axis length ratio, it would be vice versa to the description in this paper and in Lieb-Lappen et al. (2017): a value of 1 would present a perfectly isotropic object and a value of 0 an infinitely long anisotropic pore. I think therefore that the whole description of anisotropy should be checked. It is actually intuitively surprising to find the*

*highest anisotropy in the mid horizon (as the authors as well as Lieb-Lappen et al. (2017) describe), rather than near the bottom of sea ice, where brine channels and seawater are well connected. It is finally worth mentioning that anisotropy, if defined as minor to major axis ratio in this way, would be a problematic measure when considering through-sample brine channels. For this case the major axis is limited by the sample length and the measure would be size-dependent.*

We agree with the reviewer's suggestion that we should present a more accurate description of what is happening. The analysis software spits out two results, on different scales. The first is the ratio of the major axis to minor axis. This puts it on a 1 (isotropic) to infinity (anisotropic) scale. The second, and the one that we use (and was used by Lieb-Lappen et al., 2017) is the equation DA = 1 - (minor/major). This puts it on the 0 (isotropic) to 1 (anisotropic) scale. This mean intercept length method for degree of anisotropy is discussed by Odgaard, A. 1997. Three-Dimensional Methods for Quantification of Cancellous Bone Architecture. *Bone*, 20, 4. 315-328.

Our observation was that the highest DA was found in the mid-horizon, which may at first be counterintuitive. We have therefore attempted to improve our explanation. We came up with an analogy that we think is helpful (it helped us), and although it is a bit goofy, we decided to add it to the text. Bear with us:

Envision pasta. We can all agree that pasta shells (or gnocchi) would be a good model for the isotropic case. Spaghetti (pre-cooked) is clearly anisotropic. However, the spaghetti in the box is extremely anisotropic. If you dump the uncooked box on the ground, it becomes isotropic even though each individual piece is anisotropic. Brine channels in the mid-horizon are more like the spaghetti in the box. Horizontal connectivity makes it more isotropic in warmer (bottom) ice. If we were to have used an alternative definition of anisotropy and think of it as a measure of disorder/entropy, then, we agree this would be counterintuitive. However, that is not how we have defined DA for this paper.

Finally, the reviewer comments that the DA measure is limited by the length/size of sample. Yes, that is correct, which is part of the reason for NOT using the one to infinity scale. The high end of that scale is most affected by extremely long anisotropic channels, and obviously never reaching infinity. By using the formula stated above, these asymptotically approach 0, and thus, is not cause for concern here. We have added text to this section to help clarify these points.

*P 22, L424 –> Submerged cores appear to have more porous ice structure. –> Could this be supported by some of the XRT masurements? Proposing this and the following from only the photographs sounds a bit speculative.*

The JY11 no-DMP sample (red open triangle) represents ponded ice. There is only one ponded core that was scanned, and we are reluctant to make sweeping conclusions from one core due to the high spatial variability of ponded vs. non-ponded ice, but it does appear that while total porosity is similar in the ponded vs. non-ponded no-DMP JY14 cores, open pore volume at the

bottom of the core was greater and anisotropy was much lower in the ponded core, which is consistent with our field observations.

*P 22, L424 –> By June, the salinity profile shows freshening at the ice bottom, likely associated with the onset of bottom ablation. –> Another explanation could be, as the authors proposeed earlier, that this warmer ice has wider pores and looses much more brine during sampling. Fig. 9c actually supports this. If true, then the ice may only have an apparently lower salinity. This question could be addressed by a closer look into the XRT images.*
This is a good point. We have added language to the text suggesting this possibility. This is also a very good idea for future studies, to use the micro-CT imagery to address, in high spatial detail, the nuances of brine loss due to enlarged pore structure.

**4. Discussion**
*P 22, L424 –> In particular, the micro-CT work is useful for sampling much larger sample volumes, and thus central for estimating size and number distributions for the July ice. –> This claim raises several questions: 1. How may the number density of inclusions be effected by the DMP flooding process? 2. The micro-CT measurements were limited to a voxel size of 280 micron - how can optical and micro-CT number estimates be combined and compared?*

It is certainly true that the problem of DMP flooding could bias estimates of size and number distributions, however use of an optical microscope is essentially impossible once the inclusion sizes become large.

We agree it is not practical to combine and compare optical microscope estimates and micro-CT estimates. Rather, they likely dovetail each other. We hope that presenting both, along with the other datasets, adequately justifies the key finding: rotten ice is highly porous, due to pore enlargement, and therefore will behave differently than early-season ice.

*P 23, L448-451 –> Normally, sea ice with significantly smaller bulk density would be expected to float higher in the water and thus have larger freeboard. But the density reductions that occur during advanced melt result from large void spaces within the ice that are typically in connection with the ocean. As a result, such ice can have small freeboard, even if total ice thickness is still relatively large. –> I would interpret the low densities rather due to rapid brine drainage during sampling, creating apparent low densities. This question should be further adressed. Again, the micro-CT observations may be used here for clarification, by splitting them up into brine, air and DMP porosities.*
Clearly the cores sampled on this ice drain significantly when removed from the ice. But we think that the liquid that drains out is not brine, per se, but rather is seawater in free exchange with the ocean, and that it is consistently flushed through the ice.  This then becomes a semantic argument… is brine only the liquid that is trapped within the ice? Or does it apply to ocean water that invades the ice? So, likely this discussion is about open vs. closed structures. The micro-CT evidence suggests that very little closed pore volume remains in rotten ice, and that connectivity is very high in rotten ice. Visually, we saw evidence of large channels through which seawater could penetrate deep into (fully through?) rotten ice. We view this as an excellent topic for future research.

*P 24, L488-491 –> Our findings are consistent with those of Jones et al. (2012), which used cross-borehole DC resistivity tomography to observe increasing anisotropy of brine structure during spring warming. In that work, the brine phase was found to be connected both vertically and horizontally and the dimensions of vertically oriented brine channels gradually increased as the ice warmed. –> I agree, this is consistent, and it is what one intuitively would expect. However, in the results section (P 20, L378-388) you say something different. This again underlines the above mentioned inconsistency in the anisotropy description from Lieb-Lappen et al. (2017).*

Yes, we see how this would have been confusing. We have added text to clarify this, as the Jones et al. study pertains to the spring warming transition (April – June) and did not observe ice later in the summer, as in this study.

*P 24, L492-493 –> As you have results from microscopy and micro-CT you could quantify this results. E.g. plot both size distributions in a histogram. This would indicate to what degree the methods are comparable in the overlapping regime, and what resolution a CT-Scanner should have.*

The reviewer makes a very good point here. It seems this would be an excellent study to carry out in future work, where care is taken to treat microscopy samples and micro-CT samples identically and to assess how the two measurement techniques overlap—where they align and where they differ.

*P 25, L518-522 –> As mentioned above, the drainage in the lab would produce 'rotten' ice with a lot of air voids, while in the field ice may 'rot' differently, increasing mostly the brine porosity. Could you comment on the question, to what degree the applied model treats air and brine scattering differently?*

This is an important point. The following text has been added: "Differences between ice rotted in air and floating in the ocean would likely be the rate of rot, and the relative abundance of gas-filled pore space relative to liquid pore space. Refractive index contrasts mean that gas pores scatter more effectively than brine filled pores; thus, lab-rotted samples were flooded in order to best mimic in situ rotted ice."

**5. Discussion**
*P 26, L538-542 –> See above note: I would interpret the low densities rather due to rapid brine drainage during sampling, creating apparent low densities.*

This is an interesting question. Traditional density measurements may need to be re-considered for rotten ice. It is not clear how to even define bulk sea ice density in the case of ice with such highly-connected pore space. One distinction would be whether the liquid that occupies the pores is in free exchange with the ocean water, or whether it is actually in freezing equilibrium with the ice. The measurements we report here are strictly for the mass-per-unit-volume of samples that have been extracted from the ice cover. This does not determine how the ice floats / the location of freeboard.

*P 26, L548 –> critical difference –> In terms of....scattering?*

Critical difference in the way that heat is delivered to the ice. This phrase has been added.

**III Figures and References**
Thank you again for the detailed feedback here, which clarifies our presentation.

*Fig. 6 –> It would be nice to have the measured freeboard indicated in the different profiles.*
The position of freeboard was added to Figure 7 (was Figure 6).

*Also an easy-to-see distinguishment of ponded and unponded ie would be helpful.*
Ponded ice is now indicated by open circles, with "normal" ice as closed/filled markers.

*P 25, L520 –> Fig. 10, dashed curve –> the 'dashed' is difficult to see*
We made the dash larger.